

# Error in hydraulic head and gradient time-series measurements: a quantitative appraisal

Gabriel C. Rau[1,2], Vincent E. A. Post[3], Margaret A. Shanafield[4], Torsten Krekeler[3], Eddie W. Banks[4], and Philipp Blum[1]

[1]Karlsruhe Institute of Technology (KIT), Institute of Applied Geosciences (AGW), Karlsruhe, Germany
[2]Connected Waters Initiative Research Centre (CWI), The University of New South Wales (UNSW), Sydney, Australia
[3]Federal Institute for Geosciences and Natural Resources (BGR), Hannover, Germany
[4]National Centre for Groundwater Research and Training (NCGRT) and College of Science and Engineering, Flinders University, Adelaide, Australia

**Correspondence:** Gabriel C. Rau (gabriel.rau@kit.edu)

**Abstract.** Hydraulic head and gradient measurements underpin practically all investigations in hydro(geo)logy. There is sufficient information in the literature to suggest that head measurement errors may be so large that flow directions can not be inferred reliably, and that their magnitude can have as great an effect on the uncertainty of flow rates as the hydraulic conductivity. Yet, educational text books contain limited content regarding measurement techniques and studies rarely report on
measurement errors. The objective of our study is to review currently-accepted standard operating procedures in hydrological research and to determine the smallest head gradients that can be resolved. To this aim, we first systematically investigate the systematic and random measurements errors involved in collecting time series information on hydraulic head at a given location: (1) geospatial position, (2) point of head, (3) depth to water, and (4) water level time series. Then, by propagating the random errors, we find that with current standard practice, horizontal head gradients $< 10^{-4}$ are resolvable at distances
$\gtrapprox 170$ m. Further, it takes extraordinary effort to measure hydraulic head gradients $< 10^{-3}$ over distances $< 10$ m. In reality, accuracy will be worse than our theoretical estimates because of the many possible systematic errors. Regional flow on a scale of kilometres or more can be inferred with current best-practice methods, but processes such as vertical flow within an aquifer cannot be determined until more accurate and precise measurement methods are developed. Finally, we offer a concise set of recommendations for water level, hydraulic head and gradient time series measurements. We anticipate that our work
contributes to progressing the quality of head time series data in the hydro(geo)logical sciences, and provides a starting point for the development of universal measurement protocols for water level data collection.



## 1 Introduction

Water level and hydraulic head time series are critical for understanding water flow-related processes and properties in both surface and subsurface aquatic environments. At the surface, water levels are important to understand relationships between water level and flow, and for estimating surface water-groundwater interactions (e.g., Kalbus et al., 2006; McCallum et al., 2014). In the subsurface, measurements of hydraulic head are used to determine groundwater flow, estimate aquifer properties, and investigate aquifer processes such as the response to pumping or groundwater recharge (e.g., Freeze and Cherry, 1979; Domenico and Schwartz, 1997). While it has been confirmed in several studies that the accuracy of water level measurements is a limiting factor for drawing conclusions about hydro(geo)logical processes (e.g., Saines, 1981; Silliman and Mantz, 2000; Devlin and McElwee, 2007), measurement errors are not always properly recognised (Post and von Asmuth, 2013).

Pressure transducers (PTs) have been used since the 1960s to measure water level (Liu and Higgins, 2015), and the development and availability of a wide variety of commercial instruments has made collection of high temporal resolution water level time series common practice. This has been a major advancement of our capability to study hydro(geo)logical processes, but the proper use of automated sensors means that researchers need to have a good understanding of instrument technology and operating procedures. This is by no means trivial, and certainly much more complex than collecting manual measurements. Knowledge is already required during the procurement phase, as there are many brands and logger types available, and the specific research objectives of a project determine which sensors are suitable and which are not (Dunnicliff and Green, 1993). The same is true for modern positioning and levelling instruments, needed to establish the horizontal and vertical position of the monitoring well (e.g., Brinker, 1995; Hegarty, 2017). The storage and quality assurance of the large volume of time-series data is not straightforward either, and can require programming skills to process data in an efficient manner. All things considered, modern hydrologists and hydrogeologists require a broad skill set, which is typically too extensive to be comprehensively covered in standard textbooks and water-related educational programs.

Yet, water level measurement lies at the heart of all hydro(geo)logical investigation and knowledge of the measurement error associated with modern instruments is fundamental to the collection of reliable time series data. Studies on the topic published in the literature focus mainly on the instruments themselves. One of the first estimates of PT drift was published by Rosenberry (1990), who showed how these errors would have led to incorrect interpretation of water levels at several sites. More recently, Sorensen and Butcher (2011) examined the accuracy and drift of different brands of PTs and found that the manufacturers' specifications were not met during field deployment. The effect of temperature on sensor performance has also received some attention (Cuevas et al., 2010; McLaughlin and Cohen, 2011; Liu and Higgins, 2015). These studies concluded that strong temperature fluctuations such as those that occur under field conditions affect PTs of all types.

More comprehensive treatments of the subject tend to be published as reports by national research organisations. Prime examples include Freeman et al. (2004) and Cunningham and Schalk (2016) who not only discussed sensor technology, but also provided technical procedures for collecting water levels and some of the errors involved. Moreover, some relevant works were published in a non-English language (e.g. Bouma et al., 2012; Ritzema et al., 2012; Morgenschweis, 2018) or as conference





proceedings (e.g. Atwood and Lamb, 1987; Simeoni, 2012; Mäkinen and Orvomaa, 2015) so that, despite their usefulness, their findings did not permeate the indexed international literature.

When collecting hydraulic head time series in the field, many different factors apart from instrument drift influence the stability of the measurement setup (Post and von Asmuth, 2013). These include cable stretch, well clogging, sensor fouling,
variable-density effects and even changes in the vertical position of the observation well. This requires regular field site maintenance, re-calibration, and record-keeping. However, without knowledge of the magnitude of the water level error caused by such effects, there is no general guidance to develop adequate systematic field procedures. Unrecognised and unaccounted for systematic errors can accumulate, leading to inaccuracies, while the random errors increase the uncertainty. Sweet et al. (1990) contended that the propagation of measurement errors can result in $\pm 100\%$ uncertainty in calculated flow velocities, and that
the uncertainty of the head gradient may be of a similar magnitude as that of the hydraulic conductivity.

While the large uncertainty of head gradients due to water level measurement error has also been confirmed by others (Silliman and Mantz, 2000; Devlin and McElwee, 2007), there is currently no single resource that ties the lessons learned during decades of experience together. The objective of the present paper is to address this gap by quantifying the smallest possible head gradients that can be resolved using currently-accepted standard operating procedures in hydrological research.
Using data collected in a wide range of field settings we provide a comprehensive and quantitative analysis of the systematic and random errors that must be considered when collecting water level time series using automated instruments. The emphasis is on transient effects and errors that can change with time. We further add to the existing literature by highlighting sources of error that are generally overlooked. Furthermore, we propagate the random errors to quantify the best-possible composite uncertainty of horizontal and vertical head gradients considering error magnitudes from good field practice and a wider spatial extent than
Silliman and Mantz (2000) and Devlin and McElwee (2007). We anticipate that our analysis is helpful to field practitioners at all levels, and can be used as an educational resource. By providing a concise list of best practice recommendations at the end of the manuscript, we intend to provide a starting point for the development of comprehensive and universal international standard procedures, which are currently lacking.

## 2 Review of measurements and error terminology

### 25  2.1 From measurements to heads

In this work we use the term *groundwater monitoring infrastructure* (GMI) as an umbrella term for open and cased boreholes, wells, standpipe or grouted-in piezometers (Section 4). The most typical GMI in hydrogeology consists of boreholes equipped with a standpipe piezometer, where the standing water level is in contact with the atmosphere (open GMI, see location 1 in Figure 1) and therefore readily accessible for measurement. Fully grouted-in piezometers contain a single or multi-array string
of PTs and are closed to the atmosphere (closed GMI, see location 2 in Figure 1) and are often used in mining and geotechnical engineering (e.g., McKenna, 1995; Mikkelsen and Green, 2003).





**Figure 1.** Overview of the four individual measurements (enumerated as steps and marked in red) required to calculate time series of hydraulic head (one location) and gradient (two locations) using two different types of groundwater monitoring infrastructure (GMI). Location 1 shows a cased borehole that is open to the atmosphere (open GMI), whereas Location 2 illustrates a fully grouted-in piezometer (closed GMI). The boreholes are drawn at an angle to highlight the importance of errors caused by inclination during construction of the borehole.

GMI allows access to measuring *depth to water* or *groundwater pressure* from which the *hydraulic head* can be calculated (terminology is illustrated in Figure 1). The hydraulic head is defined as (e.g., Freeze and Cherry, 1979)

$$h(x,y,z,t) = z_h(x,y) + \frac{p(x,y,z,t) - p_b(x,y,z,t)}{\rho(x,y,z,t)g}$$

$$= z_h(x,y) + h_p(x,y,z,t) \tag{1}$$

where $(x,y,z)$ are the Cartesian coordinates [m] of the measurement point, $t$ is time [s], $z_h$ is referred to as elevation head [m],
5 $p$ is the total groundwater pressure [Pa], and $p_b$ is the barometric pressure [Pa], $\rho$ is the groundwater density [kg m$^{-3}$] across the water column above $z_h$, $g$ is the gravitational constant [$\approx 9.81$ m s$^{-2}$]. The term $h_p$ [m] is the *pressure head*.





The dependence of the variables in Equation 1 on $(x, y, z, t)$ has been deliberately emphasised to stress the point that their magnitude varies in space and time. Determining hydraulic head time series requires four measurements (hereafter also referred to as steps), which are conceptualised in Figure 1 and can be summarised as follows:

(1) Geo- or relative positioning of the GMI, i.e. determining its location at the Earth's surface $s_g = (x_g, y_g, z_g)$ (Section 3);

(2) Establishing the point of (or location representative for) head measurement $s_h = (x_h, y_h, z_h) = s_g + \Delta s_p$ with $\Delta s_p = (\Delta x_p, \Delta y_p, \Delta z_p)$ being the vector that represents the location offset $s_h$ with respect to $s_g$ (Section 4);

(3) Measurement of the water depth below the top of casing $d_w(t_j)$ at a discrete times $t_j$ (open GMI only, Section 5);

(4) Automated pressure measurements at PT location $s_{pt} = (x_{pt}, y_{pt}, z_{pt})$ of $p_{pt}(s_{pt}, t_i)$ at discrete times $t_i$ (Section 6).

There are two methods to obtain $h(x_h, y_h, z_h, t) = h(s_h, t)$ based on field measurements:

**Method 1 (only for open GMI):** When only $d_w$ has been measured in the field (for example, by taking regular manual water level measurements) the hydraulic head simply follows from

$$h(s_h, t_j) = z_g - d_w(t_j), \tag{2}$$

where $t_j$ is the distinct time at which the water level measurement was made.

Hydraulic head time series are nowadays commonly determined from the pressure readings of a transducer located at elevation $z_{pt}$ [m]. The head is then calculated using

$$h(s_h, t_i) = z_g - d_w(t_j)$$
$$+ [h_{pt}(s_{pt}, t_i) - h_{pt}(s_{pt}, t_j)], \tag{3}$$

where $h_{pt}$ is the transducer pressure head, i.e., the pressure recorded by the PT expressed as a water column height (e.g., Hölting and Coldewey, 2013)

$$h_{pt}(s_{pt}, t_i) = \frac{p_{pt,abs}(s_{pt}, t_i) - p_b(s_b, t_i)}{\overline{\rho}_w(t_i) g}$$
$$= \frac{p_{pt}(s_{pt}, t_i)}{\overline{\rho}_w(t_i) g}, \tag{4}$$

where $p_{pt,abs}$ and $p_{pt}$ are, respectively, the absolute and relative transducer recorded pressures and $\overline{\rho}_w$ is the average density [kg m$^{-3}$] across the water column above the transducer's elevation $z_{pt}$. Application of Equation 4 is referred to as *barometric compensation*. The location of the barometric pressure measurement $s_b = (x_b, y_b, z_b)$ must be chosen so that it is representative for the barometric pressure experienced by the PT (Post and von Asmuth, 2013).

**Method 2 (for open and closed GMI):** For a PT installed at location $s_{pt} = s_h$, the hydraulic head follows from

$$h(s_h, t_i) = z_{pt} + \frac{p_{pt}(s_{pt}, t_i)}{\overline{\rho}_w(t_i) g} = z_{pt} + h_{pt}(s_{pt}, t_i). \tag{5}$$

This is the only way by which heads can be measured in closed GMI for which $d_w$ can not be determined.





For open GMI $\overline{\rho}_w$ can be measured. For closed GMI, however, $\overline{\rho}_w$ is the average density of the groundwater above $z_{pt}$, which has to be estimated in the absence of direct measurements. Because the PT is at elevation $z_{pt} = z_h$, $p_{pt} = p - p_b$, and Equation 5 is identical to Equation 1 when $\overline{\rho}_w = \rho$. These considerations have important implications when density effects influence the pressure-head relationship of a GMI (Section 6.4.2 ).

## 2.2 Hydraulic head gradient

Hydraulic head is a scalar quantity and the gradient of the head field in combination with hydraulic conductivity enables quantification of groundwater flow rates using *Darcy's Law*. In three dimensions the hydraulic head gradient (or simply head gradient) is a vector defined as (e.g., Domenico and Schwartz, 1997)

$$\nabla h = \mathbf{i}\frac{\partial h}{\partial x} + \mathbf{j}\frac{\partial h}{\partial y} + \mathbf{k}\frac{\partial h}{\partial z}, \tag{6}$$

where the bold $\mathbf{i}$, $\mathbf{j}$ and $\mathbf{k}$ symbols denote the unit vectors in the $x$, $y$, and $z$ direction, respectively. Since $h$ and $\nabla h$ are continuous field variables, and, in practice, $h$ can only be measured at discrete points $s_h$, head measurements can only be used to approximate $\nabla h$. Moreover, it is rare for field studies to determine $\nabla h$ in three dimensions. Therefore, for the purpose of error propagation (Section 7) we consider the horizontal (in the $x-y$ plane) and vertical components (indicated by either a superscript $h$ or $v$, respectively) separately by

$$\left(\frac{dh}{ds}\right)^{h,v} \approx \frac{\Delta h}{\Delta s_h^{h,v}}, \tag{7}$$

where the term on the left-hand side represents the rate of head change per unit of distance $s$, which is approximated by the ratio of $\Delta h$, the head difference between two points of measurement, over

$$\Delta s_h^h = \sqrt{\Delta x_h^2 + \Delta y_h^2} \tag{8}$$

or

$$\Delta s_h^v = \Delta z_h, \tag{9}$$

where $\Delta x_h$, $\Delta y_h$ and $\Delta z_h$ are the distances between two points of head measurement in the $x$, $y$ and $z$ direction, respectively.

It must be emphasised that considering the horizontal head difference between two points is only meaningful when they are located along the direction of maximum rate of head change, i.e. perpendicular to the contour planes of equal head (assuming isotropic and constant-density conditions). Hydraulic head measurements from at least three different locations are required to determine the head gradient in two dimensions (e.g., Freeze and Cherry, 1979), or four locations in three dimensions (Silliman and Mantz, 2000; Devlin and McElwee, 2007). Even more locations are required for head contour maps (e.g., Ohmer et al., 2017).

## 2.3 Barometric effects

The following discussion is only applicable to open GMI (i.e., open to the atmosphere; location 1 in Figure 1). Air pressure changes act differently on the water column in open GMI than on the groundwater, because in the open GMI the air pressure





change is transmitted instantaneously to the water, whereas the groundwater pressure response is more complex and can be delayed. Barometric pressure can change as part of the local weather (e.g., the passing of high and low pressure systems) by as much as 1.5 m water level equivalent for the extremest weather events. If a barometric pressure change propagates through the unsaturated zone of an unconfined system without delay, the water level in an open GMI is a direct representation of the

groundwater pressure. However, since the unsaturated zone can resist air movement, for example under low (air) permeability or variably saturated conditions (e.g., Weeks, 1979), there can be a time lag between barometric pressure changes and the associated GMI water level response (e.g., Rasmussen and Crawford, 1997). This can be quantified using the barometric response function, which can change over time (Rasmussen and Crawford, 1997; Spane, 2002; Butler et al., 2011).

In addition to this, the response to air pressure changes of an open GMI's water level is fundamentally different than the

response of the hydraulic head due to the elastic storage behaviour of the subsurface. This can be understood by considering that an increase in barometric pressure raises the total stress acting on both the GMI's water column and the subsurface. The additional stress is borne exclusively by the water column inside the GMI, whereas it is shared between the water and the formation in the surrounding subsurface (e.g., Freeze and Cherry, 1979; Domenico and Schwartz, 1997). As a result, the pressure increase inside the GMI is larger than the groundwater pressure increase, which induces water flow from the GMI into

the formation, thus leading to a lowering of the measured water level. The result is an inverse relationship between changes in water level inside open GMI and the changing barometric pressure (e.g., Meinzer, 1939; Gonthier, 2003). This relationship can be exploited to detect aquifer confinement (Acworth et al., 2017), but also necessitates the correction of water levels measured in open GMI to faithfully infer the hydraulic head in the formation.

The barometric efficiency ($BE$) expresses the ratio between the water level change in a GMI $\Delta h_{pt}$ and the barometric

pressure change $\Delta p_b$ causing it (Jacob, 1940; Clark, 1967; van der Kamp and Gale, 1983)

$$BE = -\frac{\Delta h_{pt}}{\Delta p_b}\overline{\rho}_w g = \frac{\Delta h}{\Delta p_b}\overline{\rho}_w g = \frac{n\beta}{n\beta + \alpha}. \tag{10}$$

where $n$ is the total porosity of the formation [-], $\beta$ is the compressibility of water ($\approx 4.59 \cdot 10^{-10}\ Pa^{-1}$) and $\alpha$ is the (undrained) compressibility of the formation [$Pa^{-1}$]. The minus sign is due to the discussed inverse relationship between $h_{pt}$ and $p_a$.

The BE quantifies the partitioning of the total stress change between the formation and the groundwater (Domenico and

Schwartz, 1997; Acworth et al., 2016a). If the subsurface is assumed to be incompressible ($\alpha = 0$ so $BE = 1$, an often-made assumption), the inverse relationship between water level measured in the GMI and hydraulic head in the subsurface is most pronounced. However, the majority of geological materials are more compressible than water ($\beta > \alpha$), so realistically $0 < BE < 1$ (Rau et al., 2018). Methods to reduce barometric effects on hydraulic head measurements were suggested in the literature and are referred to as *barometric correction* (not to be confused with barometric compensation, Equation 4) (e.g.,

Hubbell et al., 2004; Toll and Rasmussen, 2007; Noorduijn et al., 2015). This discussion highlights that the $BE$ of a formation is an important property, and ignoring it can have significant implications, when hydraulic heads or gradients are derived from water level measurements with the aim to interpret groundwater processes (Spane, 2002).

Avoiding barometric effects require GMI with a specific design. Hubbell et al. (2004) suggested a sealed well and showed that their design reduced barometric pressure effects by an order of magnitude, especially for sites with deep vadose zones.





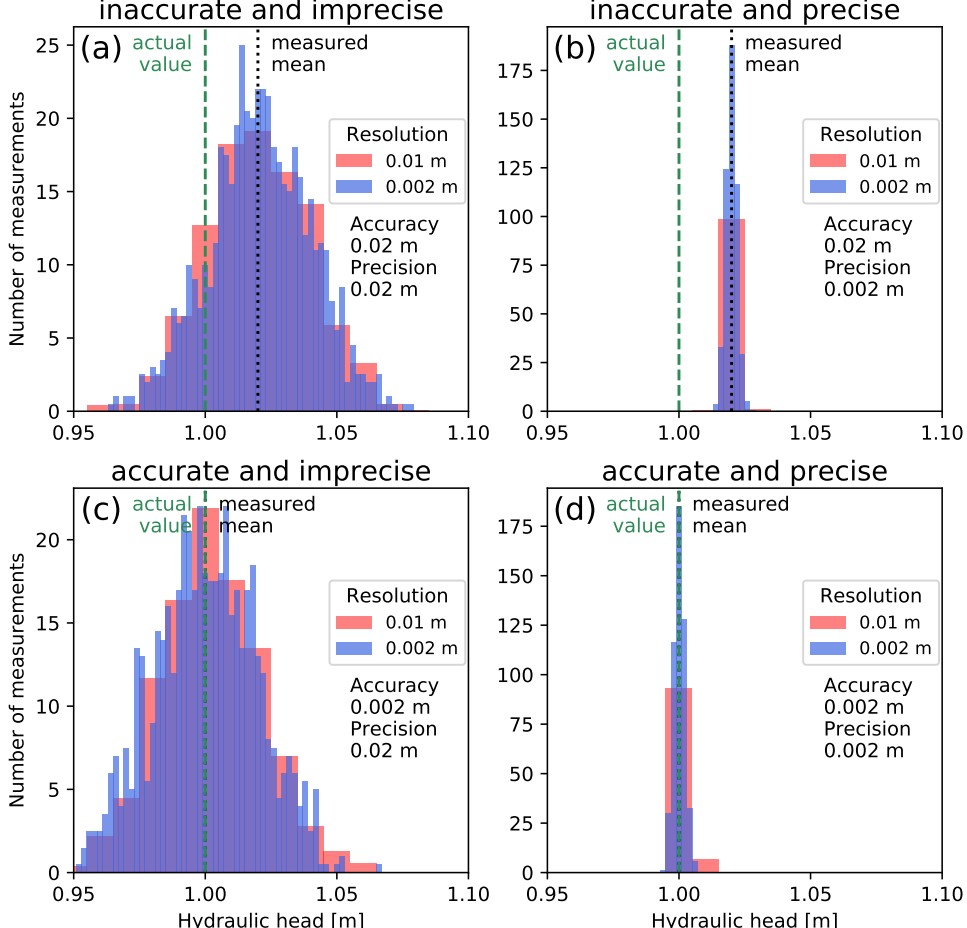

**Figure 2.** Possible combinations of accuracy, precision and resolution illustrated in a matrix, when 1,000 measurements of the same head ($h = 1$ m) are made. Measurements are (a) inaccurate and imprecise, (b) inaccurate and precise, (c) accurate and imprecise, (d) accurate and precise. Examples are illustrated with two different values of accuracy, precision and resolutions (equal to bin width in histogram).

Furthermore, a laboratory study by Noorduijn et al. (2015) demonstrated that measured total pressure recorded in sealed and unsealed wells are equal assuming barometric pressure is also measured, water levels can be accurately measured in either sealed or unsealed standpipes. This is convenient for fluvial environments, where long standpipes are subject to the forces of river flows, which can be quite violent in ephemeral streams especially (e.g., Shanafield and Cook, 2014).

## 2.4 Clarification of error terminology

Despite their importance, the terms related to measurement error are often mixed up or used ambiguously. Thus, before proceeding, it is crucial to clarify their meanings within the context of head measurement.





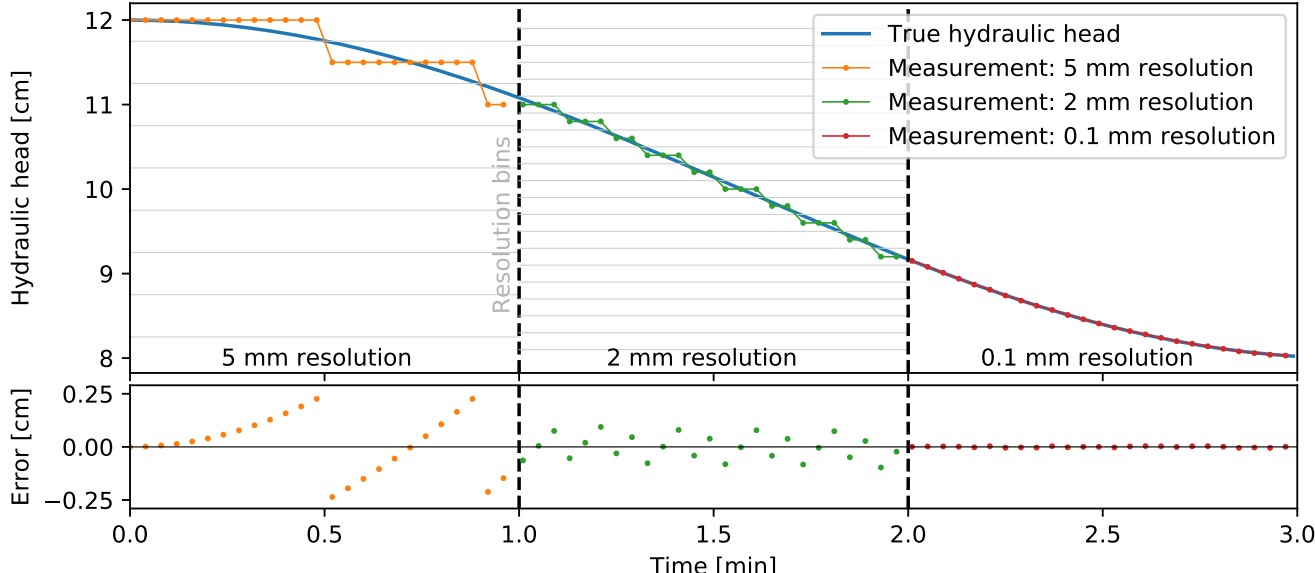

**Figure 3.** Illustration of the influence that the instrument resolution has on the measurement error: A continuous, time variable head is measured at discrete time intervals by instruments with analogue to digital conversion resolution of 5 mm, 2 mm and 0.1 mm.

**Accuracy** is a measure of how closely the mean of the measured head corresponds to the real head. The deviation between the true value and the mean of its measurements is the *systematic* (or *absolute*) error (Figure 2).

**Precision** is the *spread* of the measured heads around their mean value. When the measurements are normally distributed, it can be expressed by the standard deviation of a Gaussian distribution. It is also referred to as *random* error (Figure 2).

5 **Resolution** is the smallest numerical separation at which the change of real value can be distinguished.

**Range** is the difference between the minimum and maximum value an instrument can measure.

Electronic measurements use analogue-to-digital converters (ADC), which convert continuous analogue signals into discrete (digital) values. ADCs generally have limited steps (resolution bins, Figure 3), leading to an inverse relationship between the measurement range and resolution. Consequently, the larger the range of measurement, the coarser the resolution. For example, 10 a 12-bit ADC has $2^{12} = 4,096$ resolution bins, which equates to a theoretical resolution of 2.4 mm, when the range is 10 m, or a resolution of 12.2 mm, when the range is 50 m. As Figure 3 demonstrates, the difference between continuous and instrument-reported (quantised) head, and thus the measurement error, decreases with increasing resolution.



## 3 Geo-spatial positioning of groundwater monitoring infrastructures

There are two ways to determine a GMI's position ($s_g$, Figure 1). The first is surveying, which is the determining of the three-dimensional distance between points of interest. The second is to use navigation satellites. This section briefly summarises both. More details on surveying can be found in Brinker (1995), and on satellite system technology and applications in Hegarty
(2017), Bock and Melgar (2016) and Misra and Enge (2010).

### 3.1 Relative positioning using traditional surveying

Determining the horizontal and vertical distances to a reference point (known as trigonometric levelling) can be done using a total station theodolite. They are equipped with a precision telescope that can rotate in the horizontal and vertical direction, allowing visual adjustment of the telescope to points of interest. Precise optical sensors can pinpoint a bar-code on the staff
and digitise the angle and azimuth readings from which the horizontal and vertical distances are calculated using a built-in computer. They further include an electronic distance measurement (EDM) device, based on the travel time of laser pulses reflecting of a target, and have satellite receivers to determine geo-coordinates (Section 3).

Levelling, the technique of measuring vertical distances (heights) relative to a known survey benchmark, can be conducted using optical or light-based instruments operating from a tripod. The latest generations of optical levelling instruments use a
rotating precision telescope to magnify the scale printed onto a levelling rod (staff) that is held vertically on top of a point of interest. The telescope is used to read the vertical distance above the point of interest of a laser beam rotating in a horizontal plane. The levelling rod is equipped with a receiver that can be moved vertically until it detects the beam.

The maximum measurement distance of digital levels or total stations is limited to hundreds of meters, depending on the telescope, the range of the laser beam and the visibility of the target (El-Ashmawy, 2014). Longer distances are surveyed by
leap-frogging survey devices along multiple points (traversing) (Brinker, 1995). Measurement error is a function of distance and accuracy and precision of leap-frog surveys tend to be poorer than surveys where the instrument does not require moving. When GMI locations are to be referenced with respect to a national datum, the accuracy is further dependent on the quality of the known benchmarks that provide the link between the local survey to the national datum (Figure 1]).

It is difficult to determine the accuracy of high-precision surveying, because this must be compared to a more accurate bench-
mark method. The measurement error for state-of-the-art survey devices depends on many factors, including instrument setup, calibration, sun position, temperature elevation gradient, battery level, and most importantly, operator's expertise (Beshr and Abo Elnaga, 2011; Bitelli et al., 2018). The literature contains very few peer-reviewed investigations that test manufacturers' specifications. However, one assessment has illustrated that digital levelling can reach an accuracy of 2 mm/km with precision of 1 mm + 1 mm/km, respectively (Bitelli et al., 2018). Leap-frogging using 150 m distance steps found an elevation precision
of 1.9 mm $/\sqrt{km}$ (Ceylan and Baykal, 2006).

Estimating the positioning errors of total stations is even more complicated due to the combination of EDM and angle sensors (Walker and Awange, 2018). Braun et al. (2015) thoroughly investigated the accuracy and precision of industry standard EDM devices over a well-calibrated distance of 40 m. They found that the accuracy varied from 0-4 mm, with some devices showing




dependence on the measurement distance. We use the precision of 0.5 mm stated by Braun et al. (2015) for our error analysis (Table 1).

## 3.2 Navigation satellite positioning

Global Navigation Satellite Systems (GNSS) currently available include the widely-used Global Positioning System (GPS; USA) and Globalnaya Navigazionnaya Sputnikovaya Sistema or Global Navigation Satellite System (GLONASS; Russia), as well as Galileo (European Union) and BeiDou (China), which are currently being deployed. Additionally there are local systems such as Indian Regional Navigation Satellite System (IRNSS; India) and the Quasi-Zenith Satellite System (QZSS; Japan). Each system type consists of a network of satellites that orbit the Earth at between 18,000 - 25,000 km altitude.

The network satellites transmit their location and absolute, synchronised time, encoded in radio signals with at least two different frequencies. A GNSS receiver can decode these signals and calculate the distance to multiple satellites using the signal arrival times. In the case of global systems, the intersect of distances from at least four individual satellites enables a GNSS device to calculate location in geo-coordinates via trilateration. Single-Point Positioning (SPP) requires only one GNSS receiver (Hegarty, 2017). The horizontal positioning accuracy is at best within 5-8.5 m (Zandbergen and Barbeau, 2011) and the vertical accuracy is poorer still. This is because the visible satellites are more closely aligned in a horizontal plane and the Earth shields the signals from remaining satellites, which would provide more vertical information. Recent developments have focused on eliminating the need for multiple GNSS receivers and speeding up the time required to achieve accurate positioning (Kouba et al., 2017).

Measuring locations relies on a reference system (georeferencing) that is Earth-centered Earth-fixed. A catalogue of 3D positions is given by the International Terrestrial Reference Frame (ITRF). The latter falls to within ±1 m of the World Geodetic System 1984 (WGS84) and is therefore used as the common reference frame for geo-positioning (Bock and Melgar, 2016). The *International Hydrographic Organisation* mandates the use of WGS84 as the horizontal reference for hydrographic mapping (Rizos, 2017).

Geo-positioning is based on the geographical coordinate system, which delivers the spherical coordinates latitude, longitude and height (geoidal geometry as global reference point). Measuring lengths and areas in spherical coordinates is not straightforward. For the purpose of hydro(geo)logical investigations, these coordinate points are transformed into a projected coordinate system, a 2D representation of the Earth's surface. Although, there is some uncertainty as to the origin of this projection (Buchroithner and Pfahlbusch, 2017), the most commonly used projection is the Universal Transverse Mercator (UTM) system, which divides the Earth into 60 zones and 20 latitude bands. Each zone is then assumed to be planar and coordinates are expressed in meters as Northing, Easting and Elevation (projected from the geoid to a flat surface with local zone as the reference point). Note that height (vertical distance above the ground surface) and elevation (vertical distance above sea level) should not be confused.

Differential global navigation satellite system (DGNSS) positioning can provide much better accuracy and precision than GNSS. This approach requires at least two GNSS receivers, one of which is stationary and located at a known point (base station). The base station uses single-point positioning in conjunction with its known location to calculate an error correction.



The second, mobile GNSS receiver (rover) uses the GNSS signals in conjunction with the error correction to calculate its distance from the base station. The error correction is determined from signal phase observations at both stations (Remondi, 1985). This can be achieved offline by post-processing the stored satellite signals in both receivers, or in real-time through a radio link between the rover and the base.

Recent developments in many countries have resulted in continuous operating reference stations (CORS) at strategic locations, whose error corrections can be accessed via mobile data networks, as long as there is network coverage. The most sophisticated GNSS devices can nowadays provide positioning with millimetre horizontal and sub-centimetre vertical accuracy (Li et al., 2015; Siejka, 2018). However, these innovations have yet to make it into commercial receivers. More typically, best-achievable horizontal accuracy and precision are 15 mm and 10 mm, respectively, whereas vertical accuracy and precision

are 30 mm and 40 mm, respectively (Garrido et al., 2011). These numbers have been adopted for the purpose of our error analysis (Table 1). Interestingly, Kim Sun and Gibbings (2005) found that accuracy and precision did not show any dependence on the distance to the base station within their test area of about 11 km. It should be noted that these accuracies are achievable only when there is a sufficient number of visible satellites (for both receivers). When points of interest are near or under vegetation, the geo-positioning accuracy is significantly degraded (Bakuła et al., 2009).

When traditional surveying is undertaken there appears to be a horizontal or vertical distance dependent error whereas for DGNSS this is not the case (Table 1). Using the random error estimates, a horizontal cut-off distance at which the precision from state-of-the-art DGNSS is better than that of a total station theodolite is $\approx 700$ m. For vertical distances, DGNSS become more precise than digital levelling when two locations are further apart than $\approx 15$ km in the horizontal direction. In this case it is meaningless to derive vertical head gradients. Consequently, the surveying approach should be chosen according to the

distance between the locations.

## 4   Point of head measurement

### 4.1   Representative point of measurement

For a grouted-in piezometer (Location 2 in Figure 1), the measured pressure reflects the groundwater pressure at the vertical position of the sensor (Simeoni, 2012), and therefore this represents a true point measurement. By contrast, the water column

in a GMI that is open to the atmosphere (Location 1 in Figure 1) equilibrates to the vertical groundwater pressure distribution along the subsurface screen. The mid-point of the screen is often selected as the representative point for the measurement. However, the appropriateness of this assumption has to be considered on a case by case basis. Vertical head gradients in an aquifer tend to be small under natural (i.e., not pumped) conditions, often less than $10^{-3}$. Having a lower resistance to flow than the surrounding aquifer, a piezometer provides a flow conduit (Freeze and Cherry, 1979; Elçi et al., 2003). These associated

flow head losses are very small, thus the head within a piezometer is constant. Outside of the piezometer, the total head change in the aquifer along its screen depends on the screen length. For example, for a 2 m screen the head varies by no more than 2 mm for the quoted vertical head gradient in the aquifer. This can be taken as an indication of the maximum head error for a typical piezometer in an aquifer caused by uncertainty about the elevation of the point of measurement.



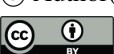

**Table 1.** Summary of precisions for the four different steps and methods required to calculate hydraulic heads and gradients. For a graphical explanation see Figure 1. Values are best possible estimates or collated from the literature.

| Measurement | Step | Option | Method | Hor. Precision | Vert. Precision | Comment/Reference |
|---|---|---|---|---|---|---|
| Geo-position of GMI (Section 3) | 1 | A | DGNSS positioning | 15 mm | 40 mm | Using state-of-the-art DGNSS systems (Garrido et al., 2011) |
| | | B | Digital total station | 1 mm + 1 mm/50 m | $1.9\ \mathrm{mm}/\sqrt{km}$ | Horizontal (Braun et al., 2015), vertical (Ceylan and Baykal, 2006) |
| | | C | Digital levelling | - | 1 mm + 1 mm/km | Vertical only, error dependent on horizontal distance (Bitelli et al., 2018) |
| Point of head (Section 4) | 2 | A | Driller's record | - | ≈ 500 mm | This is a crude estimate |
| | | B | Assuming verticality | 872 mm | 38 mm | Based on 10 m borehole length and a 5° inclination (Section 4.2) |
| | | C | Down-hole camera | - | 20 mm | Precision of winch and visual detection of screen dimensions |
| | | D | Verticality log | 87 mm | - | Based on 10 m depth and 0.5° inclination precision |
| Depth to water (Section 5) | 3 | A | Dip meter | - | 8.4 mm | Based on 256 manual measurements (Knotters et al., 2013) |
| | | B | Chalk on steel tape | - | 2 mm | Assuming twice the resolution of standard metric steel tape (Cunningham and Schalk, 2016) |
| Automated water level (Section 6) | 4 | A | Vented | - | 1.5 mm | Based on 15 months of data from 3 PTs (Section 6.2) |
| | | B | Non-vented/baro | - | 3 mm | Precision is doubled due to the use of 2 devices |
| | | C | Vibrating wire | - | 7 mm | Standard deviation of best sensor during stability test (Zarriello, 1995) |
| | | D | Laser-based | - | 0.5 mm | As tested by Benjamin and Kaplan (2017) |





However, larger errors can be expected when vertical gradients are higher than the example value used, which may be the case near groundwater discharge zones, under pumped conditions, and in formations of low permeability. Rowe and Nadarajah (1994) found that for aquitard hydraulic conductivity tests, where the propagation of an induced head drop in a piezometer is recorded as a function of time, the representative point of measurement was biased towards the bottom of the screen, and that

this significantly influenced the outcomes of the parameters to be determined. Moreover, as the gradients changed in time, so did the representative point of measurement. In layered aquifer systems, the water level in wells with long screens was found to depend on the transmissivities of the layers intersected by the wells (Sokol, 1963). These findings highlight the need for using short screens. However, the finite screen length of standpipe piezometers means that some uncertainty remains about the representative vertical position of the head measurement.

## 4.2 Borehole verticality and screen location

Despite best efforts in many countries on the mandatory requirement to report accurate information on the drilling and completion of GMIs, construction details are often reported at a precision of decimetres (not centimetres) and prone to significant systematic error. Due to the variety of different field and environmental conditions as well as the different qualifications and experience of drillers, we assume that the vertical screen locations can be estimated from driller's logs with a precision no

better than about 0.5 m (Table 1).

The deviation from vertical of a GMI further results in uncertainty about $s_h$ (NUDLC, 2011). The importance of borehole deviation surveys is critical in other industries such as oil and gas, where errors in the observed inclination angle and other parameters of the monitored fracture-system geometry impact on the monitoring and interpretation of hydraulic fracturing (Bulant et al., 2007). Yet, in hydrogeology borehole verticality is typically ignored in particular when calculating heads gradients.

Poorly aligned boreholes impact significantly on the integrity of casing, and hence increase the risk of flow short-circuiting and water column density-stratification (Section 6.4.2).

A thorough investigation of borehole deviation was conducted by Twining (2016), who applied a correction factor to water level data when a borehole deviation survey indicated a change of more than 0.06 m between the measured borehole length and the true vertical depth. From the 177 boreholes surveyed, correction factors to the historical water levels of these wells

ranged from 0.06 to 1.8 m and inclination angles from 1.6 to 16 degrees. A comprehensive examination of borehole deviation was conducted in more than 100 boreholes drilled (up to 1,000 m deep) at the Swedish nuclear repository site by the Swedish Nuclear Fuel and Waste Management Company (SKB) (Nilsson and Nissen, 2007). Their investigation provided an uncertainty of deviation measurement of the inclination of the borehole (up to 3 degrees) as well as an elevation uncertainty at the bottom of the borehole (up to 15 m for the boreholes measured).

Guidelines for drilling and water bore construction for plumbness and straightness is generally a *'do the best you can'* approach within practical limits using appropriate equipment and drilling operation (e.g. drilling centralisers, correct collar and feed pressure) for the geological conditions (NUDLC, 2011; Treskatis, 2006; BDA, 2017). Drillers consider angles of less than 5° as acceptable (Bulant et al., 2007). Hence, the horizontal positioning error of a 10 m long borehole would become:





$sin(5°)·10 \, \text{m} \approx 872$ mm, whereas the vertical error is: $[1-cos(5°)]·10 \, \text{m} \approx 38$ mm. In the absence of more literature reporting on borehole deviations, we use this reported figure as the random error when determining the point of head (Table 1).

Since this uncertainty is much greater than the achievable accuracy of the GMI's geo-position, $s_g$, the verticality (plumbness) of a borehole should be measured using downhole geophysical tools such as verticality probes or inclinometers. This includes an gyroscope or an accelerometer to measure the vertical angle combined with a magnetometer to provide the probe's rotational position around the vertical axis. Both measurements can be logged continuously while lowering the probe. For example, assuming an industry standard precision of 0.5° (e.g., Verticality Sonde by GeoVista, UK) an otherwise straight 10 m deep borehole, the resulting precision in identifying the horizontal screen offset would be $\approx 87$ mm which is an improvement over the crude guess based on a 5° angle according to best drilling practice. The borehole deviation survey should be combined with a down-hole camera to determine the position of the screen relative to the top of the GMI. We estimate the depth measurement precision of a typical system to be approximately 20 mm (Table 1).

## 5 Depth-to-water measurements

There are a number of different ways to measure depth to water ($d_w$, Equation 3). It is commonly done by hand and involves the use of a measurement tape (Nielsen and Nielsen, 2006). Most groundwater projects today use electric water level meters colloquially called *dip meters*, which provide an audible or visible signal when a sensor touches the water surface. When the acoustic signal is not electronic but an audible noise is generated mechanically, for example by lifting and dropping a hollow brass cylinder just touching the water surface, the instrument is called a *plopper*. Another inexpensive method uses a steel tape that is covered with chalk (Cunningham and Schalk, 2016).

Depth-to-water measurements should be performed frequently (at a minimum every 3 months) for checking the performance, and adjustment, of automatic sensors. Good-quality measuring tapes are marked every 1 mm (metric) or every 0.01 ft (imperial). The chalked-tape method can potentially deliver a precision that corresponds to the resolution of the graduated steel tape (Nielsen and Nielsen, 2006) (Table 1 and Figure 2), whereas dip meters and ploppers are generally read to the nearest centimetre. This may involve human measurement errors such as the switching of digits (e.g., noting 57 instead of 75) or reading to the wrong decimetre/metre marking on the tape (Knotters et al., 2013).

There is only minor information in the literature about the errors associated with manual head measurements. The lack of assessment is surprising given that manual measurement is the most important link which ties automated pressure time-series to a benchmark (Equation 3). Some controlled experiments have been conducted though, most recently in the Netherlands by Knotters et al. (2013). Sixteen operators, with varying degrees of experience, each took a reading in a total of 16 standpipes. Half of the readings were done with an electronic dip meter, the other half with a plopper. After discarding the obvious mistakes from the data set, the errors were fitted to a normal distribution with a mean and standard deviation of 5 mm and ±8.4 mm, respectively (Table 1) for the electronic dip meter, versus 0.3 mm and ±9.5 mm for the plopper. The measurements by Knotters et al. (2013) were representative for very shallow water tables. A poorer precision (0.05 ft = 15 mm) was reported by Atwood and Lamb (1987) (cited in Sweet et al. (1990)) for water levels more than 120 m below the surface measured by




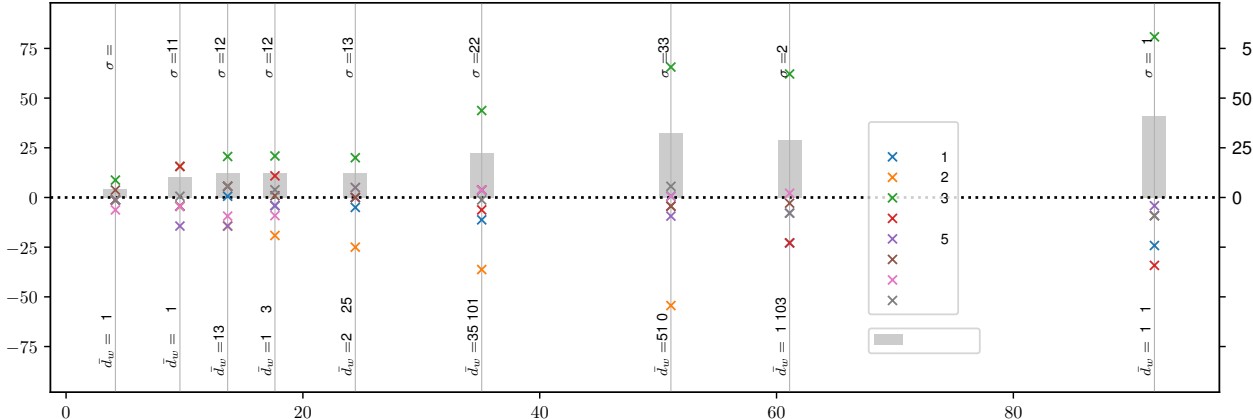

**Figure 4.** Manual measurement of 9 different groundwater depths using 8 different dip meters: Difference from mean and standard deviation (precision) over mean depth.

two observers using the same instrument within a short time period. Sweet et al. (1990) conducted a rather comprehensive experiment themselves but reported the errors as percentages which makes the figures difficult to compare to the other studies.

Knotters et al. (2013) noted that the graduations on some of the tapes that were used showed noticeable differences and that this caused systematic measurement error. Plazak (1994) compared three water level probes to a reference probe and found

differences that increased with depth, reaching a maximum value of 0.1 ft = 0.03 m at a depth of 61 ft = 19 m. Comparable findings based on our own experiments are shown in Figure 4, which summarises variations in manual measurements using several commercial electric dip meters of various lengths to measure water levels at 9 depths between 5 and 90 m. Electric water level loggers by the same manufacturer differed by several centimetres, with differences of up to 0.12 m observed overall (one person taking the measurements sequentially for each borehole, using the same location at the top of casing for each bore).

The differences increased with depth to the water table for several instruments, confirming the observations made by Plazak (1994) 25 years ago. Discrepancies of this magnitude preclude the use of data for accurately identifying small head gradients and call for replacement of the measuring instrument.

Wear (e.g. kinks and tears) on electric water level tapes causes additional discrepancies in measurements over time. Cunningham and Schalk (2016) detail procedures for calibrating electric water level devices before each use; this involves measuring

the electric tape against a steel measuring tape kept in the office only for this purpose. For consistent time series, it is extremely important that a datum on the casing is used to always measure water level from the same point (Nielsen and Nielsen, 2006; Cunningham and Schalk, 2016). This seems obvious but there can be confusion when different operators are involved, and regular maintenance is required to make sure the mark stays visible. Repeating the measurement a number of times can avoid tape reading errors and ensures proper functioning of the electronic dip meter, which may give inconsistent readings some-



times (with saltwater, for example). In areas prone to vertical land surface movement (i.e., those that are tectonically active or experience land subsidence) a regular check of the elevation of the casing ($z_g$) is necessary.

## 6 Automated water level time-series measurement

### 6.1 Automated measurements

Automated measurement of water levels or pressures in GMI requires electronic devices capable of time keeping and sensing. Many commercial instruments combine a stand-alone clock, a sensor, an ADC unit, memory and a power supply in a single housing. There are also instruments that house only the sensor and are connected to a data logger that converts the sensor signal and stores the quantised readings. The focus of this section is on systematic errors that occur during automated collection of water level data in the field.

Automated instruments have the capacity to record for a long period of time unimpeded. Data loss as a result of logger chip or battery failure can be prevented by the automated transmission of instrument readings to a receiving data management system via radio, infrared signals, a GSM network or satellites (Morgenschweis, 2018; Bailey, 2003). This is referred to as telemetry. The expansion of cellular network provider coverage and the reduced cost of data-only plans in the recent past have made telemetry systems a more accessible and viable option for remote hydro(geo)logical monitoring. Nowadays, transmitted 15 data is stored on network servers which can be accessed in real time via computer or smart phones. Telemetry systems allow remote modification of sensor settings and identification of sensor problems, early identification of logger failure as a safeguard against data loss, re-synchronisation of time keeping to avoid time based errors (Section 6.6). However, the deployment of a telemetered system does not avoid certain errors such as sensor drift (Section 6.4.3). Consequently, to ensure the accuracy of automated water level measurements, frequent site inspections and manual measurements are still required.

### 6.2 Types of devices

At present, water level time series are typically determined using pressure measured using submersible or grouted-in PTs (Figure 1) and rely on Equations 3 and 4 to determine the water level. A comprehensive overview of PTs can be found in Freeman et al. (2004) or Hölting and Coldewey (2013).

The most popular type of PT consists of a piezo-resistive crystal made from silicone or ceramics, which acts as a strain 25 gauge as it deforms under pressure. The deformation causes the electrical resistance of a Wheatstone bridge to change, which is gauged by recording the changing voltage due to a constant current. *Vented PTs* are connected to a venting tube that connects the air chamber of the submerged PT to the atmosphere. The measured pressure is the relative pressure ($p_{pt}$ in Equation 4) and there is no need for barometric compensation. We estimate a best possible precision for vented PTs to be 1.5 mm (Table 1). This value is based on water level time series recorded with three vented PTs (with a range of either 10 or 20 m) inside 30 a standpipe piezometer in Hannover (Germany) over a period of more than 15 months. Using one logger as a reference and calculating the difference with the remaining two loggers resulted in standard deviations of 1.4 and 1.7 mm ($n = 11,279$),





thereby demonstrating excellent performance. However, readings can be influenced when the venting tube does not remain dry. A desiccant capsule is therefore attached to the tube and this can causes some practical difficulties when measuring in river beds or other areas subject to flooding, as well as when freezing occurs (Liu and Higgins, 2015).

*Non-vented* PTs measure absolute pressure $p_{pt,abs}$ and are converted to relative pressure $p_{pt}$ by subtracting the barometric
pressure $p_b$ (Equation 4, barometric compensation), typically measured with a barometric PT nearby the GMI. The subtraction of barometric pressure from absolute pressure results in a loss of water level measurement precision, because the two instrument measurement errors accumulate (Section 7). Because of this, we estimate the best possible water level measurement precision as 3 mm (double that of vented PTs; Table 1).

A second type of PT is the so-called vibrating wire piezometer (VWP) that uses electromagnetic coils to excite a wire
exposed to differing strain resulting from pressure changes. The square of the resonant frequency is linearly proportional to the pressure (Zarriello, 1995). VWPs are designed for long-term stability and are therefore used for closed GMI when the instrument is fully grouted-in (Figure 1b). However, re-calibration becomes impossible once installed (Contreras et al., 2007). VWPs of the non-vented and vented type exist, and some models have a pressure range of 10 MPa (equivalent to about 1 km of water) or sometimes higher. We estimate the best possible water level measurement precision of VWPs to 7 mm (Table 1).

There are also electronic water level measurement devices that emit a laser pulse and determine the depth of water from the time it takes the pulse to reach the water table and return to the sensor (known as LiDAR: light detection and ranging). When connected to a time-keeping data logger, this technology is suitable for time-series collection (Benjamin and Kaplan, 2017). A recent development and test of a LiDAR-based system has demonstrated an outstanding precision of 0.5 mm (Table 1; Benjamin and Kaplan, 2017), but condensation and, in groundwater studies, borehole non-verticality, can interfere with the
light reaching the water surface.

Another type of water level sensor is based on electronic capacitance measurement. It consists of two electrically isolated plates or wires that are aligned in parallel at close proximity. Submergence of the wires in water creates a contrast in electrical capacitance compared to air, with values that are proportional to the submerged length. Their range (typically 1 to 2 m of water level) is smaller than piezo-resistive PTs (which can be used in water depths of 100 m or even more). An important
advantage of capacitance probes is that they are rugged and can withstand overload, drying and freezing. In contrast to the measurement-tape and LiDAR technique, which measure $d_w$ from the top downward, the capacitance probe sits in the water column and records water levels on a data logger (similar to PTs).

## 6.3   Instrument range and resolution

In their instrument specifications, manufacturers typically provide the accuracy of a PT as a percentage of the full-scale (FS)
range. Unfortunately, this number is not defined unambiguously. Typically, it may comprise a combination of a sensor's non-linearity (the relationship between $p$ and voltage $V$ not being a straight line), hysteresis (differing $p$-$V$ relations during $p$ increases or decreases) and repeatability (the closeness of measured $p$ values for the same $V$), and thermal artefacts (influence of temperature on the $p$-$V$ relation). These are non-adjustable errors and are therefore not related to accuracy in the sense that they can be corrected by applying a simple offset to calibrate the instrument to a known value. Moreover, since there







**Figure 5.** (a) Depth to water measured by three different instrument types, illustrating the influence of precision and resolution on head time series: Capacitance PT (blue line, high precision and low resolution), vented PT (orange line, high precision and high resolution), non-vented PT (corrected for barometric pressure) (green line, low precision and high resolution). The examples are from (a) Ti Tree Basin in the Northern Territory (Australia) and (b) a farm dam in South Australia (note the effect of a 0.2 mm rainfall event on 5 January, which is visible in the orange line but not in the green line).

are different ways to quantify the instrument's deviation from the ideal $p$-$V$ relation, the number specified as the instrument's accuracy can have a different meaning depending on a manufacturer's definition. This can even mean that an instrument with 0.5% FS range accuracy is as accurate as an instrument with 0.1% FS range accuracy (STS Sensors, 2017).

As a practical example of precision and resolution (see Section 2.4), Figures 5a and 5b show several days of automated depth 5 to water level measurements made using different logger types. The difference between the graphs is the vertical scale, with the



water levels in Figure 5a fluctuating at the mm scale and those in Figure 5b showing an overall decline of a few centimetres. The curves recorded by different water level measurement devices illustrate the discrete time and magnitude nature of automated measurement. Here, the blue line illustrates high precision and low resolution values, the orange line shows high precision and high resolution measurements, whereas the green line represents low resolution and low precision data.

Current standard practice allows the quantification of subsurface processes and properties from time changes in heads due to either natural or induced causes. Common examples such as aquifer tests rely on large head changes that are sufficiently resolved using off-the-shelf instruments. However, more recent research advances have demonstrated that subtle signals in hydraulic heads can also be used to passively quantify hydrogeological processes and properties. For example, Figure 5 demonstrates sub-centimetre diel (i.e., daily) fluctuations that originate from phreatophyte evapotranspiration (e.g. Gribovszki et al.,

2013) or Earth and atmospheric tides (e.g. Acworth et al., 2015). Such subtle signals can only be detected with appropriately high sensor resolution. Currently, it is advisable to deploy vented transducers to minimise errors resulting from clock differences and precisions due to barometric compensation (refer to Section 2.1). For such intentions the measurement range must be minimised in favour of maximum resolution, which reduces the measurement error (Figure 3).

## 6.4    Issues related to pressure transducers

### 6.4.1    Temperature effects

The response of piezo-resistive sensors to pressure changes is a function of temperature, hence most instruments record temperature alongside pressure and use this to compensate the readings. Nevertheless, the operation of PTs in transient temperature environments has been found to affect water levels that are calculated from pressure readings. For example, Cain et al. (2004) showed that when PTs are exposed to direct sunlight, thermal effects add noise to water level measurements. Sorensen and

Butcher (2011) noted that temperature compensation often significantly compromises the accuracy of pressure readings.

For non-vented PTs, it is especially important to consider placement of the barometric PT to prevent adding noise from thermal effects into water levels during barometric compensation (Cuevas et al., 2010; McLaughlin and Cohen, 2011), which is demonstrated in Figure 6. The graph of water level versus time shows a clear diurnal variation in the time series recorded by a non-vented/barometric PT pair. The data were converted to water levels by subtracting the recorded barometric pressures from

the total pressure recorded by the non-vented PT, which was suspended in a surface water pond. The manual measurements taken between 3 and 6 November do not confirm the variations inferred by the logger data. Inspection of the diurnal temperature variations of the barometric PT shows daily temperature variations of 10°C or more prior to 16 November, on which date the PT was placed in a more constant temperature environment. As a result, the periodicity that characterised the water level data before that date disappears from the water level time-series. Gribovszki et al. (2013) argued that such thermal effects

should not affect vented PTs, which therefore should be suitable for fine-scale (e.g. sub-daily) measurements as required for evapotranspiration calculations. However, Liu and Higgins (2015) found that rapid changes in temperature on a sub-daily time scale can cause the air in the line to expand or contract, and that the relationship between temperature fluctuation and logger error varies between loggers.




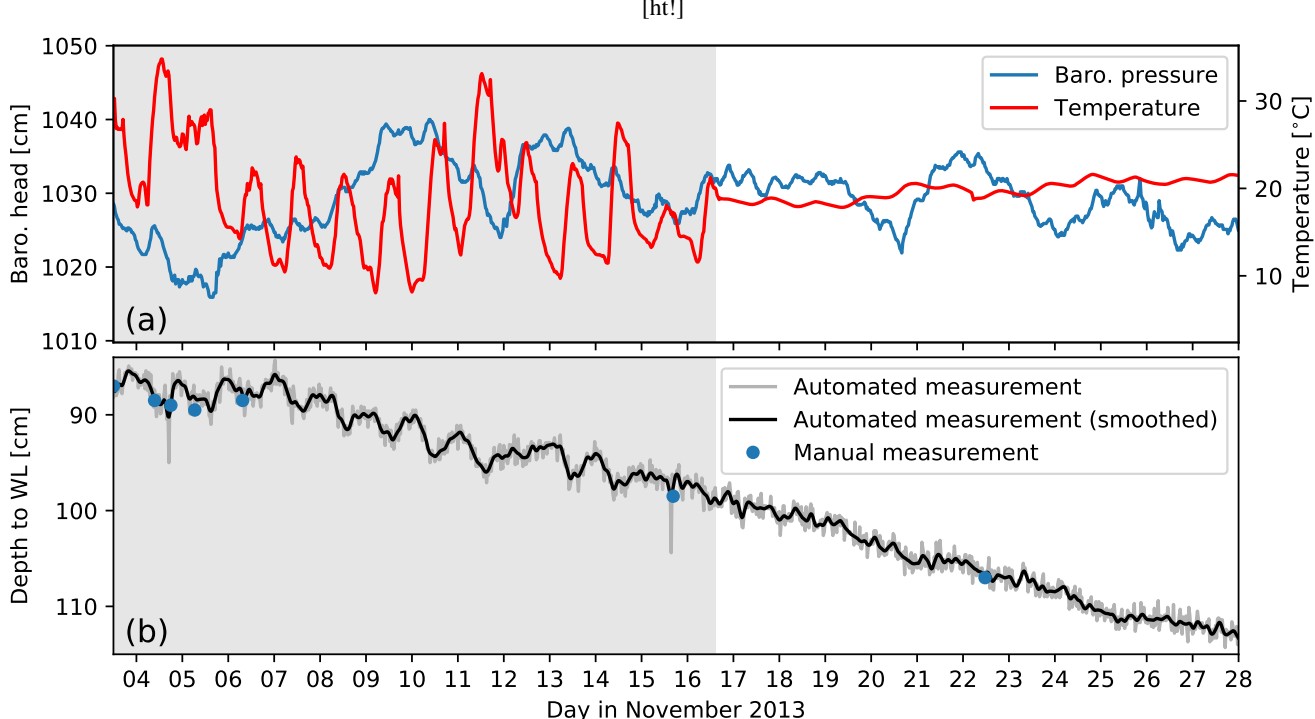

**Figure 6.** (a) Barometric pressure and temperature and (b) water level versus time. The data in (a) were collected using a logger that was exposed to significant temperature fluctuations prior to 16 November 2013. The water head time series in (b) was derived by subtracting the measured pressures shown in (a) from the total pressures measured using a non-vented PT. The artefacts caused by the temperature variations prior to 16 November 2013 are clearly reflected by diurnal oscillations of the water levels (grey shaded area). Note that the manual dips confirm that there was no diurnal variability in the water levels (blue dots).

### 6.4.2  Water column density changes

For internally consistent hydraulic head time series, it is imperative that the average density across the water column $\overline{\rho}_w$ in Equation 4 stays constant in time. Strictly speaking, this is never the case and the impact of the changes of $\overline{\rho}_w$ represent a systematic measurement error that has to be assessed and, when not negligible, corrected for. The effects are largest in groundwater systems with changes in salinity, such as coastal aquifers (Post et al., 2018). When the density varies within the water column and with time, $\overline{\rho}_w$ is given by (e.g., Post and von Asmuth, 2013)

$$\overline{\rho}_w(t) = \frac{1}{h_{pt}} \int_0^{h_{pt}} \rho(z,t)dz, \tag{11}$$

where $\rho(z,t)$ is the density as a function of the vertical dimension and time, $h_{pt}$ is the vertical distance between the top of the water column and the PT (Equation 4).



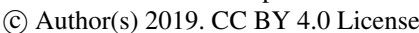

**Figure 7.** (a) Time series of water level pressure recorded by two transducers located shallow (superscript $S$) and deep (superscipt $d$) in the same well. (b) Difference between the deep and shallow pressure measurements. (c) Water level in an observation well showing offset in mean water level after the well was purged for hydrochemical sampling ($AHD$ is *Australian Height Datum*). Note that due to the high temporal resolution, the discrete measurement points are not shown and the time-series are drawn as lines instead.

Application of Equation 11 requires knowledge of the density distribution across the length of the water column at multiple times, which is seldom collected. Pressure to head conversion errors due to unknown knowledge of $\overline{\rho}_w(t)$ is therefore probably one of the most overlooked issues in head time series measurement. Some instruments provide a correction function based on the change of the electrical conductivity and temperature measured by sensors housed in the same instrument as the PT, but this is only meaningful if the density of the water column above the PT is constant. It is important to distinguish the effects





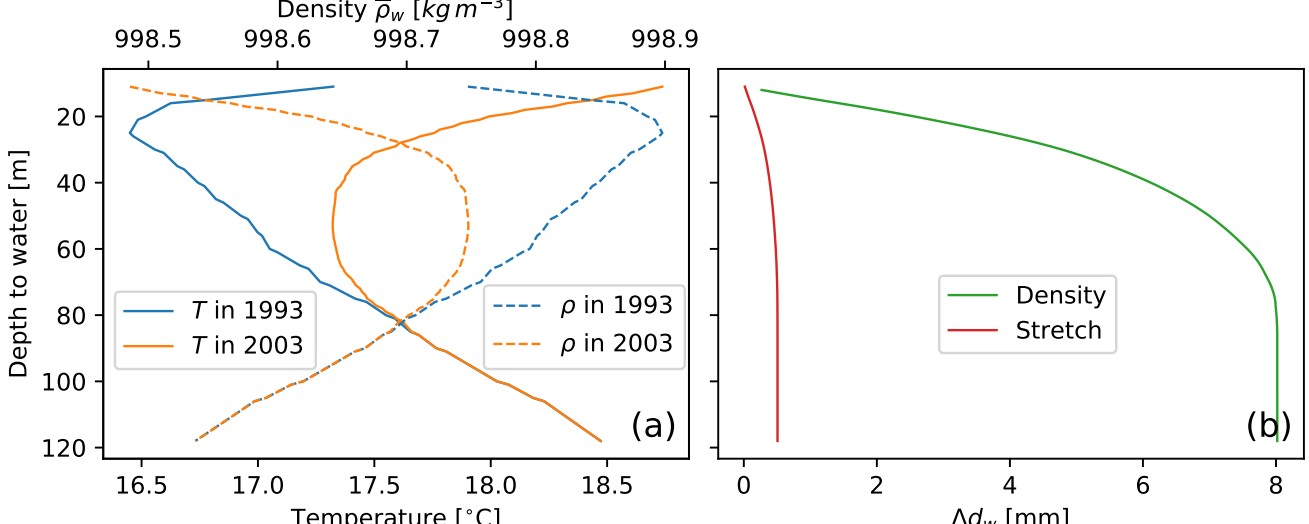

**Figure 8.** (a) Temperature and average density as a function of depth for an observation well in Japan that experienced significant warming due to urbanisation between 1993 and 2003. Temperature data from Yamano et al. (2009). (b) Theoretical head increase due to the temperature-related density decrease and head increase that would be perceived if the PT's vertical position is lowered due to thermal expansion of the steel suspension wire.

discussed here from the head corrections that must be applied when studying flow in variable-density groundwater systems (Lusczynski, 1961; Post et al., 2018).

A subtle effect of the change of $\overline{\rho}_w$ with time is shown in Figure 7. The upper graph shows the pressures recorded during an experiment whereby two PTs were hanging inside the same standpipe piezometer. One was located just beneath the air-
5    water interface, and one was just above the bottom of the piezometer. The latter case corresponds to the way the pressures are recorded when heads are calculated using Method 2 (Section 2.1), whereas the former is representative for Method 1. Because of the well's vicinity to the sea, the recorded pressures vary with the tide. The difference between the PT readings is shown in Figure 7b. In a well with a constant $\overline{\rho}_w$, the pressure difference would be constant in time. Clearly, this is not the case here and two effects are notable: (i) a linear trend (grey dashed line in Figure 7b), causing the pressure difference to become smaller and
10    (ii) oscillations that are superimposed on this linear trend.

The linear trend was due to leaking casing joints, which led to the ingress of fresh groundwater in the upper parts of the piezometer, as a result of which a salinity stratification developed. As more freshwater seeped in with time, $\overline{\rho}_w$ decreased by an amount $\Delta\overline{\rho}_w$ per unit of time $\Delta t$. In fact, the slope of the linear trend line is equal to $\Delta\overline{\rho}_w g h_{pt}/\Delta t = 200$ Pa d$^{-1}$, which for $h_{pt} \approx 50$ m gives $\Delta\overline{\rho}_w/\Delta t = 0.04$ kg m$^{-3}$ d$^{-1}$, and this is roughly consistent with the estimate of $\Delta\overline{\rho}_w/\Delta t = 0.03$
15    kg m$^{-3}$ d$^{-1}$ derived from consecutive downhole probe measurements on 1 July and 5 August 2015. The superimposed tidal oscillations were caused by the change of the density stratification inside the piezometer standpipe with the tide: as the tide rose,





groundwater with an ambient, high salinity entered across the well screen and this caused more saltwater to stand above the deepest PT. The shallow PT, however, remained in the freshwater part of the stratified water column. Both PTs experienced the same increase in water column height above the sensor, but because this added height consisted of freshwater for the shallow PT, it sensed a smaller pressure change than the deeper PT. Correcting for these effects shows that the pressure difference

becomes virtually constant although not all fluctuations disappear. The fluctuations around the mean difference decrease to become around 0.1 kPa (1 mm of water column height). The cause of the remaining fluctuations is not clear; they may be due to clock synchronisation issues.

    While the previous example showed a subtle trend of relatively low magnitude, the time series in Figure 7c show the potentially large magnitude of an abrupt change of $\overline{\rho}_w$. In this case it was caused by the purging of the well for hydrochemical

sampling. Prior to sampling, the water inside the well had a non-constant salinity, because it had not been properly developed at the time of construction. After sampling, the well was filled with water with the same salinity as the groundwater at the well screen. As a result, $\overline{\rho}_w$ increased from 1006.7 to 1015.1 kg m$^{-3}$. Based on these density values, the length of the water column inside the well would have changed from 72 m to $^{(72 \cdot 1006.7)}/_{1015.1} = 71.4$ m, i.e. a decrease of 0.6 m, which corresponds to the measured change of 0.67 m; the additional difference may be due to the removal of silt and other fouling material from the

well screen by the pumping.

    An example of the effect of temperature-related density changes on the head error is shown in Figure 8. In this example, the change in temperature was caused by urbanisation, resulting in a noticeable warming of the upper 75 m of the subsurface. This caused the density of the water column to decrease, and hence a longer column of water is required to balance the pressure at the screen. Figure 8b shows the magnitude of this effect as a function of depth, which in this example remains limited to

less than one centimetre. Another effect that plays a role is the lengthening of the logger's metal suspension wire as it warms. Assuming a linear expansion coefficient for steel of $11 \cdot 10^{-6}$ $K^{-1}$, the increase in wire length as a function of depth is showing in Figure 8c. The magnitude of this effect is less than 1 mm. Because this example was chosen to represent a case of relatively strong temperature increase for groundwater, it is expected that these values represent the upper bounds for thermal expansion effects, which thereby represent relatively small errors under typical groundwater conditions. Larger effects could occur though

near aquifer thermal storage facilities, geothermal areas, or in very deep wells.

### 6.4.3   Measurement drift

Sensor drift is one of the most common errors in automated hydraulic head measurements. Here it is expressed as $\Delta d_w$, which is defined as the depth below TOC measured manually with a dip meter minus the depth measured by the PT. Sorensen and Butcher (2011) tested 14 different transducer brands commonly used in hydrogeological studies. For PTs with a range $< 15$ m

H$_2$O, the drift was observed to be $-8 \leq \Delta d_w \leq 27$ mm after 99 days in the field, but the models with a greater range showed up to 5 times more drift. Data available to the present study from Syria, where 11 observation wells were equipped with vented PTs in January 2009 and were not inspected until June 2010, showed $-199 \leq \Delta d_w \leq 153$ mm, with only one of the PTs showing a negative $\Delta d_w$ value.





In an extensive study of 473 piezometers all equipped with the same brand logger and inspected every three months for a total of two years, Pleijter et al. (2015) statistically evaluated $\Delta d_w$ values based on a data set of 5,583 measurements. For 144 piezometers, a statistically-significant linear trend could be identified. The slope of the trend line was negative for 95 and positive for 49 of the piezometers. The drift was reported (as the median of the trend line slopes) to be -3.6 cm yr$^{-1}$ and 4.4 cm yr$^{-1}$ for the negative and positive trends, respectively.

Apart from technical reasons that cause PT drift, fouling of instruments is a well-known problem that affects the quality of head time series. This can be due to the formation of mineral precipitates by hydrochemical processes (Sorensen and Butcher, 2011). Biological processes often build biofilms of microorganisms, or larger organisms such as snails attach themselves to a sensor. Biofouling filters consisting of copper coiled wire that can slow down these effects, but regular inspection and cleaning are a requirement to prevent measurements from being compromised. Moreover, improper suspension cables may stretch, hence causing the logger to sit deeper below the water surface, or frequent removal of loggers for downloading may cause the wire length to change due to kinks.

Drift introduces errors of unknown magnitude that remain unnoticed unless identified by frequent checks using an independent measurement (Rosenberry, 1990). The examples of field-observed drift in Sorensen and Butcher (2011) and Pleijter et al. (2015) show that drift is not generally linear. The rate of change can vary in time, sometimes suddenly, and even reverse direction. Frequent, independent $d_w$ measurements by manual dipping provides the only means to correct for drift. Drift correction involves removal of the linear trend between manual measurements, which introduces uncertainty because of the linearity assumption. Drift corrections must be carefully documented and at all times must the original data be stored alongside with the corrected data. Data downloaded at different times must be stored separately to ensure that a drift correction applicable to a particular block of data is not inadvertently applied to other blocks of data.

Another form of drift, which is related to the GMI and not the PT itself, occurs if the conditions in the GMI change such that the relationship between the recorded water level $h_{pt}$ in the well and the groundwater pressure $p$ in the aquifer (Equation 4) is not constant over time. When the change of $\overline{\rho}_w$ with time (Section 6.4.2) is responsible for this, it will cause $\Delta d_w \neq 0$. However, $\Delta d_w \neq 0$ cannot be used to detect measurement errors due to clogging of the well screen by suspended sediment particles, geochemical processes (e.g., iron oxidation) or biofilm growth. An example of the detrimental effect on time series because of the latter phenomenon is illustrated in Figure 9, which shows the water level in a piezometer as a function of time. The temporal dynamics remained very much subdued until the well screen was mechanically rehabilitated in June 1996 (Willemsen, 2006). As soon as the hydraulic connection between the piezometer and the aquifer was restored, the temporal variability of the head in the aquifer became apparent.

## 6.5 Clock drift

Automated water level and pressure recorders use an autonomous or external clock which relies on crystal oscillators (commonly made of quartz) for a counter measuring process that forms the basis for digital time keeping. The crystal oscillators are highly accurate, yet small deviations in their oscillation frequency, which changes with time (a phenomenon known as ageing), can add up over long measurement periods. This results in a gradual drift of the internal clock in relation to the real time.





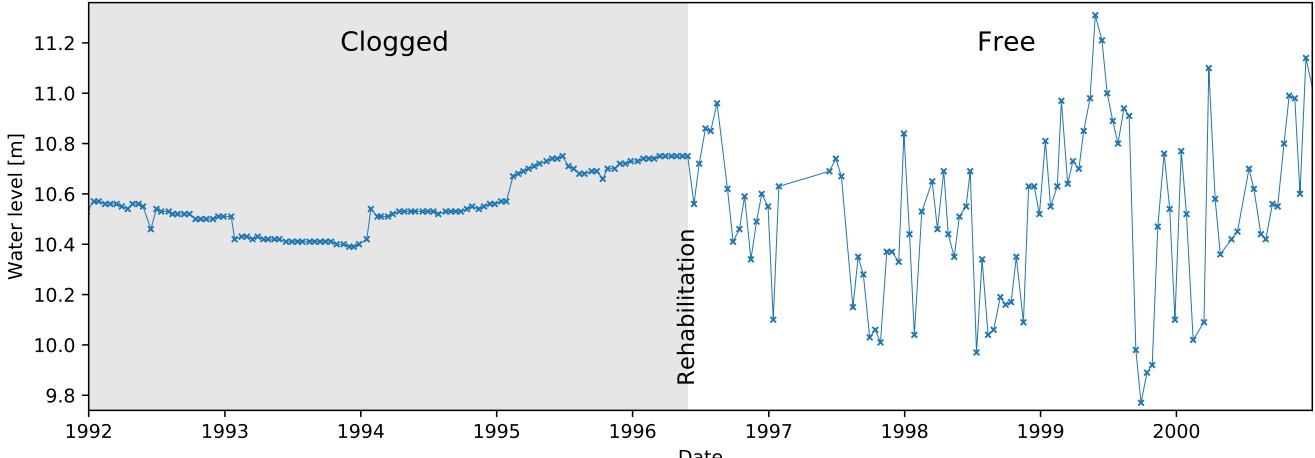

**Figure 9.** Water level versus time for a piezometer in the Netherlands that had a clogged well screen until it was rehabilitatated in June 1996 (Willemsen, 2006). The temporal dynamics of the head in the aquifer were not registered by the piezometer until that time. Data were obtained from http://www.dinoloket.nl (downloaded on 16 January 2019).

Elimination of this form of systematic error can be achieved by synchronising the transducer clock with a more accurate clock such as in a field laptop. Clock stability is an important consideration when using multiple instruments. Examples include the barometric correction of absolute pressure measurements from a non-vented transducer, or the calculation of hydraulic gradients using two different time series.

The clock stability of 8 PTs was assessed during a long-term surface water-groundwater exchange monitoring program in the arid zone of Australia (Fowlers Creek at Fowlers Gap, New South Wales, Australia), where streams are dry for most parts of the year, but flow if there is enough rainfall (Acworth et al., 2016b). Monitoring stream flow under such conditions relies on long-term and accurate resolution hydraulic gradients. To monitor the spatial and temporal dynamics of stream flow we used streambed arrays similar to those reported in McCallum et al. (2014). Before deploying the PTs, the internal clock of the field

laptop was synchronised to an online time server. This ensured that all loggers had the same time stamp. The transducers were setup to start logging on 21 October 2014 at 18:00 AEDT (*Australian Eastern Daylight Time*) with a sampling interval of 30 minutes.

Due to the remoteness of the field site, monitoring continued for over 2 years. After removal and disassembly of the streambed arrays, the internal clock of each PT was compared to that of a synchronised computer. The findings demonstrate

that the majority of the PTs did not comply with the manufacturers specifications of $\pm 1$ min $y^{-1}$, with most of the clocks running slower and the worst clock drift being +7.5 min $y^{-1}$ (Table 2).

The influence of clock stability on measuring hydraulic heads and gradients is illustrated in Figure 10. In this example, a vertical array similar to that used in McCallum et al. (2014) was deployed in a streambed at Maules Creek (New South Wales).





**Table 2.** Example of an assessment of clock stability for 8 different standard PTs (AEDT is *Australian Eastern Daylight Time*). None of the PTs complied with the clock stability of $\pm 1$ min yr$^{-1}$, as specified by the manufacturer.

| Logger serial | PT end time AEDT | Actual time AEDT | Time difference [min] | Record duration [days] | Clock drift [min yr$^{-1}$] |
|---|---|---|---|---|---|
| 2004398 | 27/05/2016 14:24 | 27/05/2016 14:32 | 8 | 584 | 5.0 |
| 2004777 | 27/05/2016 15:10 | 27/05/2016 15:21 | 11 | 584 | 6.9 |
| 2020185 | 27/05/2016 15:15 | 27/05/2016 15:11 | -4 | 584 | -2.5 |
| 2020180 | 27/05/2016 15:39 | 27/05/2016 15:51 | 12 | 584 | 7.5 |
| 2005097 | 27/05/2016 15:39 | 27/05/2016 15:41 | 2 | 584 | 1.2 |
| 2005116 | 27/05/2016 14:38 | 27/05/2016 14:50 | 12 | 584 | 7.5 |
| 2005086 | 17/11/2016 10:13 | 17/11/2016 10:24 | 11 | 758 | 5.3 |
| 2020164 | 17/11/2016 10:30 | 17/11/2016 10:36 | 6 | 758 | 2.9 |

Resolving vertical head gradients over small distances is a significant challenge. Both PTs were calibrated against each other by placing the array inside a water bath overnight.

Figure 10a shows the pressure heads as well as the vertical head gradient during the experiment. Figure 10b illustrates the outcome if either one of the PTs was synchronised at a different time or as a result of clock drift. It is clear that the largest error arises during largest head changes with time, where the gradient disregarding time errors could be interpreted as either gaining or losing conditions with different magnitudes. Similar to this example, Post et al. (2018) showed how clock drift led to erroneous flow estimates in a coastal aquifer subject to tides. Hydrological processes could be fundamentally misinterpreted if time related monitoring errors are ignored, which is not always properly recognised.

## 6.6 Miscellaneous errors

The importance of setting the logger to the appropriate time resolution (sampling rate) is illustrated by Figure 11. Both lines show the same water level time series, but the red line shows how the curve would look if the measurement frequency had been set to twice daily, whereas the blue line shows the data as measured using an interval of 30 minutes. Obviously, the short-term variations caused by the operations of a nearby production bore are not captured when an inappropriate measurement interval is chosen. Similar issues can arise in aquifers affected by tides or river stage fluctuations.

When unresolved temporal head fluctuations occur between two consecutive automated measurement intervals $t_i$, a large discrepancy can arise between a manual measurement taken at time $t_j$ and the nearest measurement at time $t_i$. When field





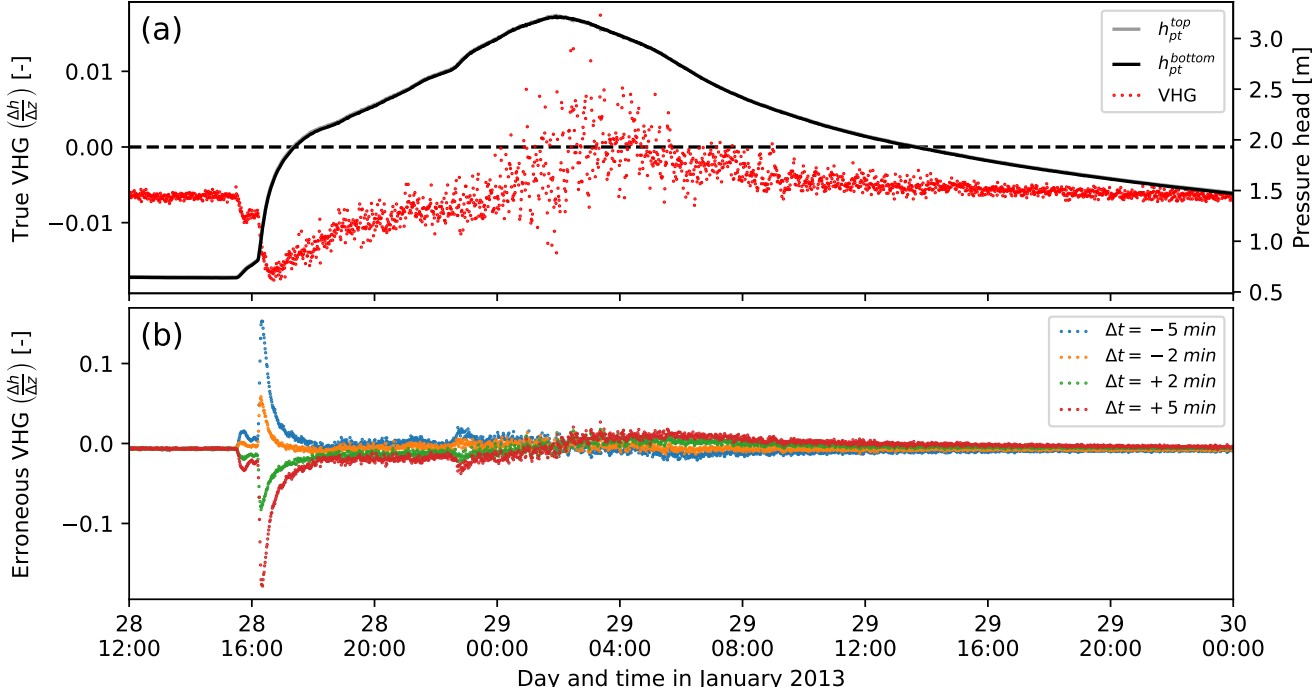

**Figure 10.** The influence of clock stability on calculating a vertical head gradient: (a) Pressure heads measured in a streambed using two PTs (note that the two lines are too close together to distinguish) with clocks that are in sync and the calculated vertical head gradient. (b) Erroneous vertical head gradients arising from time differences $\Delta t$ due to out-of-sync instrument clocks caused by clock drift (the data was synthetically shifted).

personnel take manual measurements during their regular site visits, the timing of which is usually not determined by the logger recording settings but by logistical factors instead, considerable differences can arise from the fact that $t_j \neq t_i$. Using the data in Figure 11 as an example, a manual head measurement taken at time $(t_j)$ of 09:11 on 5 September 2015 would be 0.83 m higher than the closest automated reading of the logger (set to a 12 hour sampling interval) at the time $(t_i)$ of 12:00.

5    The difference is unrelated to any instrument error and is solely due to unresolved temporal variability. A manual dip taken at any time between the two sampling times would result in an error that falls within the grey box in Figure 11. While this hypothetical example represents an extreme case of this effect, misalignment of $t_i$ and $t_j$ is very common and it supports the contention by Sweet et al. (1990) that unrecognised hydrological processes are a form of measurement noise. To avoid such errors, a measurement interval of at most 1 hour must be chosen upon initial logger deployment. Only when it becomes clear

10    that there is no temporal variability at this timescale can the sampling interval be increased.

     There is a variety of reasons why PTs do not always accurately record the water level in open GMI, many of which can be prevented by proper installation. When suspension cables are attached to well caps, the logger may not always be in exactly the same position after having been removed from the GMI. Some lightweight pressure transducers may experience buoyancy,




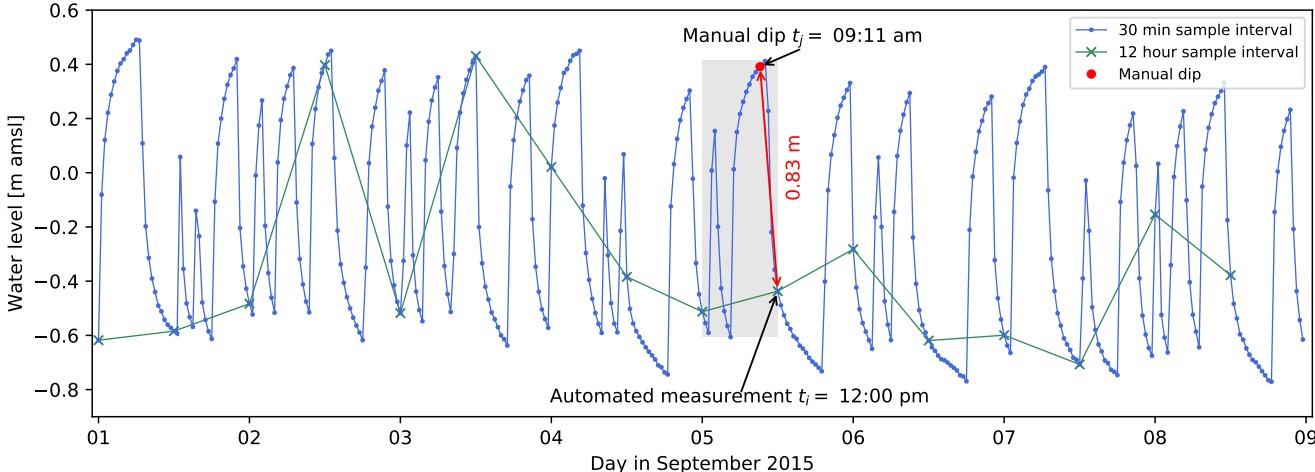

**Figure 11.** Water level in an observation well responding to irregular and frequent pumping from a nearby production bore automatically measured at two different sampling intervals (12 hours or 30 minutes). Note that a manual dip which is taken at a time that falls between samples of automated measurement (e.g., the time period indicated by the grey box) can result in a significant measurement error in dynamic hydro(geo)logical systems.

especially in saltwater, and hence their vertical position is not constant in time. As a consequence of suspension cables being too short, PTs may end up being suspended above the water surface inside the GMI when the water level falls, and hence record the atmospheric pressure (Mäkinen and Orvomaa, 2015). Air bubbles that become entrapped in the PT after the water level rises again can cause inaccurate readings and must be removed.

When open GMI becomes artesian, which can occur when water levels rise higher than the standpipe's top, the PT not longer indicates the true water level. When the PT is too deep to withstand the pressure of the water column (the so-called burst pressure, usually about twice the measurement range), the sensor may damage and the logger will malfunction. Freezing of the water column and lightning strikes can also cause damage to the PT (Freeman et al., 2004). Sometimes PTs show erratic readings for no apparent reason, which can be due to the overheating of electronics. Temperatures in the standpipe sticking up

above the land surface can easily exceed 40°C, the upper temperature threshold for correct functioning of electronic parts, due to sun exposure. Shading or ventilation measures are therefore also an important part of GMI.

    One issue not discussed in the literature is the considerable confusion that arises due to daylight saving time (DST) related clock adjustments. Perhaps this is because it is considered a trivial point, but an important one nonetheless that must be specifically addressed in the measurement protocol. Some devices automatically adjust to daylight savings time, whereas

others do not. When they do, the instrument's clock setting depends on the time of year it was set up. Some manufacturers apply DST corrections only when the recorded data are exported to a file and this depends on the computer's DST settings. The same data could therefore also end up with different time stamps if multiple computers are used.



**Figure 12.** (a) Visual comparison of horizontal and vertical random errors listed as precision values in Table 1 (note that some errors are distance dependent) for the different steps (Figure 1) and method options. Minimum achievable relative random head gradient error in the horizontal (b) and vertical (c) direction calculated using standard practice measurements (highlighted by the black frame: step 1 options A,B,C; step 2 options B,C; step 3 option A; step 4 option A). Note that HHG and VHG values with errors exceeding 100% are blanked out in (b) and (c).

## 7 Random error propagation

Figure 1 illustrates steps required to calculate a head gradient. In what follows it is assumed that all systematic errors have been eliminated from this process in a way that the only error remaining is the random error. For the sake of simplicity and in





the absence of further information, we also assume that all random errors are normally distributed and not correlated (Table 1). Furthermore, horizontal errors stated in Table 1 are isotropic, i.e. do not vary in the $x$ and $y$ directions. Note that all error quantification in this subsection assumes that standard practice consists of (see Table 1 and Figure 12a):

1. Horizontal distance measurement using a total station (step 1 option B), vertical distance measurement using digital levelling (step 1 option C). Distance errors are limited to the respective errors from DGNSS (step 1 option A). While GNSS surveying with a single receiver is useful for mapping, the precision of coordinates is not high enough to determine the distances between GMI for head gradient calculations.

2. Point of head measurement using downhole camera for vertical (step 2 option C) and verticality for horizontal error (step 2 option B) assuming a 10 m deep well;

3. Manual water level measurement using a dip meter (step 3 option A) assuming no depth dependency of the random error;

4. Automated pressure measurement using a vented transducer (step 4 option A).

Figure 12a graphically compares the random measurement error magnitudes for the different steps and methods summarised in Table 1. We stress that the adopted values reflect the absolute best case scenario from current standard field practice.

The random error associated with head differences arises from steps 1, 3 and 4 and can be expressed as

$$\delta \Delta h = 2\sqrt{(\delta z_g)^2 + (\delta d_w)^2 + (\delta h_{pt})^2}. \tag{12}$$

We use the random errors associated with standard practice in this equation to estimate the minimum achievable precision (combined random error) when calculating head differences as $\delta \Delta h = 0.017$ m. This error is somewhat higher than the findings by Devlin and McElwee (2007), but lower than the field-based values ($\delta \Delta h = 0.022$ m) reported by Post et al. (2018). Measured head differences that are smaller than this value will not allow much confidence in detecting the direction of groundwater flow. To improve this precision, approaches to reduce the achievable random errors when measuring steps 1, 3 and 4 must be found (Figure 1), likely resulting higher effort and cost than what is currently standard practice.

Using the measurements illustrated in Figure 1, the horizontal hydraulic head gradient (HHG) is calculated as

$$\nabla h^h = \frac{[z_g - d_w + h_{pt}]_2 - [z_g - d_w + h_{pt}]_1}{\Delta s_h^h}, \tag{13}$$

where the numeric subscripts depict the two locations. Analogously, the vertical hydraulic head gradient (VHG) can be determined as

$$\nabla h^v = \frac{[z_g - d_w + h_{pt}]_2 - [z_g - d_w + h_{pt}]_1}{[z_g + \Delta z_p]_2 - [z_g + \Delta z_p]_1}, \tag{14}$$

A propagation of random errors accounts for the errors involved in measuring the different variables explained in Figure 1. The relative error for the head gradients is as follows

$$\frac{\delta \nabla h^{h,v}}{|\nabla h^{h,v}|} = \sqrt{\left(\frac{\delta \Delta h}{\Delta h}\right)^2 + \left(\frac{\delta \Delta s_h^{h,v}}{\Delta s_h^{h,v}}\right)^2}, \tag{15}$$




where

$$\delta\Delta s_h^{h,v} = 2\sqrt{\left(\delta s_g^{h,v}\right)^2 + \left(\delta\Delta s_p^{h,v}\right)^2}. \qquad (16)$$

For the VHG case, the $\delta\Delta s_h^v = \delta\Delta z_h$, $\delta s_g^v = \delta z_g$ and $\delta\Delta s_p^v = \delta\Delta z_p$. The latter are the vertical positioning errors of the GMI and point of measurement resulting from an non-vertical borehole (steps 1 and 2). We use equation 15 to calculate the minimum achievable random relative error for HHGs and VHGs as a function of horizontal or vertical distance between two points of head measurement (Figures 12b and Figure 12c).

Figure 12 clearly demonstrates the relationship between HHGs or VHGs and distance between the points of head. In general, the greater the distance between screens, the smaller the relative head gradient error. For example, the random error of determining a HHG or VHG of $10^{-2}$ at a 10 m horizontal or vertical point of head distance is $\approx 17\%$ (see examples in Figure 12). Figure 12b further illustrates that measuring a HHG $< 10^{-4}$ with an error less than 100% requires a distance $\gtrsim 170$ m between points of head. VHGs of $10^{-4}$ are unresolvable within the considered maximum vertical distance of 100 m (Figure 12c). We stress that these errors are the best case scenario, as in reality there is a likeliness of additional systematic errors contained in the measurements. The errors calculated here are thus unlikely to be achieved in practice. Extraordinary effort must be put towards improving the precision of measurements when head gradients less than $10^{-2}$ are to be calculated for distances smaller than 10 m (Figure 12b). Note that in order to additionally determine the direction of the gradient, a minimum area between GMI is required (Devlin and McElwee, 2007).

## 8   Concluding remarks

Reliable water level measurements are at the core of every hydro(geo)logical investigation and the measurement error determines which processes or properties can be resolved. We have analysed unpublished and published data to quantify the best possible accuracy and precision of hydraulic head measurements using commonly available, state of the art, commercial instruments. By propagating the random errors, we find that with current standard practice, horizontal head gradients $< 10^{-4}$ are only resolvable at distances $\gtrsim 170$ m, and that it takes extraordinary effort to measure hydraulic head gradients $< 10^{-3}$ over distances $< 10$ m. However, we consider these estimates very optimistic, as they assume that systematic errors are absent or that systematic error corrections do not introduce additional error.

The magnitude of systematic errors tends to be much larger than that of random errors and hence failure to recognise systematic errors can seriously compromise the outcomes of an investigation. In part, systematic errors are due to the measurement conditions in the field, which are not easy to control and negatively affect instrument performance. But, other factors play a role too, including improper instrument use, faulty or unsuitable GMI (e.g., long well screens), and the lack of measurement protocols that pay due consideration to all sources of error. Some measurement techniques have not seen performance improvement in decades, and there does not seem to be the same quest for measurement error reduction in hydro(geo)logy as there is in other fields of science, where the smallest of dimensions are measured with ever-better accuracy and precision and advances in measurement technology are pushing the frontiers of science.



We acknowledge that the measurement error with available technology could already be sufficiently small for a lot of practical applications. Still, improved standards for water level measurement would be an important step towards better hydro(geo)logical data quality and consistency. Moreover, technological advances are necessary to enable the measurement of vertical flow within an aquifer, the subtle temporal head fluctuations related to tidal cycles. Increased sensor performance and

sensitivity would underpin new developments, such as the use of the groundwater response to Earth and atmospheric tides to characterise the degree of groundwater confinement (e.g. Butler et al., 2011; Acworth et al., 2017) and quantify compressible subsurface properties (e.g. Acworth et al., 2016a; Rau et al., 2018). Such advances highlight the need to innovate beyond standard practice to support research in the hydrogeological sciences. We believe that researchers and industry should work together and find ways to increase instrument performance.

The following list of recommendations synthesises the findings from our study and focuses on aspects that could considerably improve the current practice of hydraulic head measurement. These are:

**Elimination of systematic errors:** Our estimation of the minimum achievable random error across all measurements presumes the absence of systematic errors (Table 1, Figure 12). Not all systematic errors (e.g., sensor drift) can be eliminated, but to minimise human error, measurements should be conducted exclusively by personnel that has received

formal training. Moreover, a detailed measurement protocol must be designed and periodically evaluated, which outlines the procedures for measurement, maintenance, note keeping (using standardised field data sheets) and data storage and handling.

**Point of measurement:** Our review of the literature demonstrates that GMI can significantly deviate from vertical (Section 4). The resulting error in the point of head measurement is larger in the horizontal compared to the vertical direction (step

2 in Figure 1). While this potentially introduces one of the largest errors, it is generally ignored when calculating head gradients (Table 1 and Figure 12). If investigations necessitate the detection of small HHGs, we recommend measuring the borehole verticality using down-hole profiling tools in open GMI. The best possible precision in measuring the point of head is achieved by combining a verticality sonde with a downhole optical camera. Geophysical logging (Keys, 2017) or flow meter measurements can identify GMI construction errors and ageing issues, such as casing leaks. For open

GMI, joint interpretation of barometric pressure and water level time series is required to determine hydraulic heads (*barometric correction*, Section 2.3). For closed GMI, the point of head measurement accuracy depends on the details contained in the original drilling report if inclinometer casing is not used in the installation (McKenna, 1995; Mikkelsen and Green, 2003).

**Geo-spatial positioning:** Our error propagation analysis (Figure 12) clearly demonstrates that precise measurement of the

horizontal and vertical distances between GMI is paramount to resolve the small hydraulic gradients inherent to groundwater investigations. This is particularly important when the GMI locations are in close proximity. Geo-spatial data from single-receiver GNSS should only be used for mapping but not to calculate distances. Traditional surveying techniques deliver more precise results for horizontal distances <700 m compared to DGNSS. Vertical distances should only be cal-




culated using data from digital levelling and not DGNSS. If possible, leap-frogging the survey device should be avoided (Section 3).

**Automated head measurements:** The widespread use of automated PTs for hydraulic head and gradient measurement has perhaps led to the impression that manual measurements have become less important. Our analysis demonstrates that

regular, frequent manual water level measurement remains important as it is the only way to verify that PTs are accurately recording the correct water level (Section 6.4.3). It also improves the precision of the automated measurements by averaging out the error introduced from manual dipping (Table 1). Given the significance of manual measurement, it is surprising to note that commercially available dip meters show as much error today (Figure 4) as a quarter century ago (Plazak, 1994). Telemetry does not obviate field site visits, but only offers the convenience of not having to download

devices and the advantage of being able to detect potential problems remotely, albeit at a higher cost of installation and maintenance (e.g., data service and connection problems).

**Time related errors:** Automated transducers rely on the stability of their internal or external time base once synchronised with the clock of the device that is used to set up the logging protocol. We demonstrate that clocks can drift significantly (Section 6.5), which leads to silent measurement errors and false interpretations (Figure 10), especially for highly

dynamic systems, where uninterrupted long-term monitoring is required. In reality, the clock stability error must be doubled when non-vented PTs are used to assess barometric effects. We recommend that the clock is re-synchronised as frequently as possible or, where this is impossible, careful documentation of the device's internal clock status when monitoring is finished. Such practice is not always supported by off-the-shelf devices and the limitations of the software and device have to be trialled before deployment. Good time keeping practice also includes the use of one and the same

field laptop, which is regularly synchronised with a time server. Moreover, we recommend the use of an absolute time base, for example *Universal Time Coordinated* (UTC), to avoid systematic errors arising from daylight savings time confusion.

**Density and temperature effects:** We demonstrate that automated pressure measurement and accurate conversion into water levels necessitates knowledge of the average density of water inside the borehole ($\overline{\rho}_w$, Equation 11). Since water density

depends on the amount of dissolved substances as well as temperature, there is a need for measuring water temperature and electric conductivity across the length of the water column to establish their potential influence. If the water density inside the open GMI is not constant, the best solution is to position the PT such that it measures $p_{pt}$ at elevation $z_h$ (Method 2 in Section 2.1), although this may come at the cost of greater measurement error due to a larger PT range. Further, PT readings are often affected by temperature despite internal compensation (Section 6.4.1). While subsurface

temperatures beyond 2-3 m depth are generally roughly stable, avoiding temperature effects can be a significant problem when measuring water levels or barometric pressure at or near the surface.

**Type of pressure transducer:** The type of PT is an important consideration that should be made according to the purpose of the investigation. For general groundwater monitoring away from topographic depressions and waterways (no risk





of borehole over-topping) we recommend vented PTs. Because vented PTs measure a relative pressure instead of an absolute pressure, they have have a smaller range and do not require a separate instrument to simultaneously record the atmospheric pressure. As such, they have better accuracy, precision, and resolution. Also, there is less risk of human error and their use avoids the problems that are introduced with two PTs (i.e., sensor and clock drift). For reliably resolving

head gradients and flow direction at small vertical distances, for example when assessing surface water-groundwater interactions, we recommend the use of wet/wet differential pressure sensors (e.g., Cuthbert et al., 2011).

**Technical specifications:** In the technical specification of PTs, much of the focus is on accuracy as a percentage of the full-scale range. We noted that this value is not consistently defined between manufacturers and may contain adjustable (i.e., errors that can be corrected using manual depth-to-water measurements) as well as non-adjustable errors (hysteresis,

repeatability, non-linearity). Before purchasing, it is wise to approach manufacturers and enquire about the various technical details. Consideration must be paid to minimising the measurement range in favour of maximum possible resolution (Section 2.4). Practice has shown that PTs have high failure rates, hence reliability is also an important selection criterion.

As a final remark, documentation of measurement procedure is critical for data validation. Without any assessment of

the measurement uncertainty it is impossible to assign a quality label to the data, which severely limits their worthiness for consideration in public databases. In addition to data collection protocols, quality control procedures must be in place to ensure the reliability of the distributed water level data. The development of such procedures should be considered in future work.

*Author contributions.* The idea for this manuscript was conceived by GCR and VEAP, and they were the lead authors. Everyone contributed with ideas, data and writing. GCR coordinated the effort and produced the figures and tables.

*Competing interests.* No competing interests.

*Acknowledgements.* Some of the data used in this analysis were collected with equipment provided by the Australian Federal Government financed National Collaborative Research Infrastructure Strategy (NCRIS). Some of the groundwater data are available through the NCRIS Groundwater Database: http://groundwater.anu.edu.au. This project has received funding from the European Union's Horizon 2020 research and innovation programme under the Marie Skłodowska-Curie grant agreement No 835852. GCR acknowledges support from the Australian

NSW State Government's Research Acceleration and Attraction Program in 2018. EWB acknowledges funding support from the Australian Research Council (ARC) Linkage Project LP150100588. The authors thank Mr Nick White for collecting the data shown in Figure 4 and Mr Michael Teubner for the manual measurements shown in Figure 6.





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
