# Peer review of "Error in hydraulic head and gradient time-series measurements: a quantitative appraisal"

_Hydrology and Earth System Sciences, 2019_

## Short Comment (SC1) · 1 May 2019

Hello:

First of all, thanks for the precious recommendations for minimising the systematic errors. It's quite practical but not many people ever considered. I've three aspects to ask:

1) Change on the surface elevation: It's generally not considered for the most case, but there's a special occasion, that we recorded a surface subsidence from a 8 m GMI due to the severe drought in Thirlmere Lakes, NSW, Australia. It's been noticed because it's so visible, that without any changes on the GMI, the water level suspiciously increased for 3 days around 6 pm for 3 days. Then no more changes were monitored. Every

change lasts around 1 hour with more than 100 mm increase. I think this phenomenon might associate with temperature effects or simply aquifer changes. Not sure if there's good method to mitigate?

2) Change on the logger position from the wire connected: It's been mentioned, but not fully discussed in the main content. It mainly occurs with a vented logger and the venting cable, which has larger diameters than regular wires. Especially when the piezometer is narrow, the changes from that might be a big problem for this bore during a relatively large water recharge/pumping. That's why I personally don't recommend vented logger.

3) As some errors might offset other errors, and we already have a brief idea for some general ranges of errors, is that possible to have an universal accepted errors for the whole records?

Regards,

Fang

---

## Short Comment (SC2) · 14 May 2019

**Comments by: Jonathan Kennel, Jessica Meyer, Christopher Neville, Beth Parker**

This paper provides a good review of many of the factors that can influence hydraulic head measurements and it is nice to see increased focus on such a fundamental measurement. There are three areas in the paper where we feel further clarification is warranted: time-lag of the monitoring well; calculation of hydraulic gradient; and the comparison of vented and non-vented transducers. Our comments address relatively subtle points and not intended to detract from the overall emphasis of the paper.

[Figure]

**1  Time-lag**

It should also be emphasized that while the equivalent freshwater water level (corrections for density) or equivalent piezometric level (for shut in measurements) are often used to infer flow directions and calculate hydraulic gradients, the water level elevation in an open well is not the fundamental parameter of interest. The hydraulic head in the formation is what drives flow. A key distinction is that often a monitoring well or piezometer requires groundwater flow between the formation and monitoring well to record a change in water level surface. This flow is not instantaneous and is commonly referred to as time-lag (Hvorslev, 1951). This time-lag is different than the time-lag associated with the propagation of barometric pressure through the vadose zone mentioned in the current paper. The length of the time-lag is dependent on well-bore storage and formation properties; the water level measured from an open monitoring device (or even a grouted transducer to a much smaller extent) will incorporate the effects of antecedent changes in formation pressure. Monitoring wells do not all have the same time-lag associated with them, which also adds temporal uncertainty when comparing measurements. As monitoring frequency increases the uncertainty associated with variable time-lags will become more important and apparent.

**2  Hydraulic gradients**

The locations of the monitoring wells play a very important role in the calculation of the hydraulic gradient, not only the knowledge of their true position (x and y, z, and time coordinates) but how the wells are oriented in relation to each other. For example, it is worth mentioning that for calculating a planar hydraulic gradient from three wells in the same hydrogeologic unit the optimal arrangement is in the form of an equilateral triangle. As the locations deviate further from an equilateral triangle the uncertainty of water level measurements plays an increasingly important role in gradient calculations.

Gradients vary in space and time, so with increasing monitoring distances the spatial confidence of the calculated gradient actually declines (i.e., uncertainty around the representativeness of the gradient). We are limited by our devices but we strive to get gradients across appropriate scales. It should be mentioned in the discussion on vertical gradients that avoiding blending of distinct hydrogeologic/hydrostratigraphic units is critical to accurately calculating meaningful vertical gradients.

**3   Vented and non-vented transducers**

One of the main arguments against non-vented transducers is that you need to convert these values to an open hole water level measurement. While this may be the case when you want to compare the value to a manual measurement in a conventional piezometer, in many cases this is not necessary. For example, if you are trying to calculate a horizontal gradient between three wells in the same aquifer, and all of the transducers deployed are non-vented, there is no need to first remove the barometric component from the results if the elevations of the sensor of each transducer are known. Converting to an equivalent water level may just add uncertainty that is not necessary. Another example is for calculating barometric/loading efficiency. The main issue is related to time shift between equipment, which affects both methods similarly. Given the same transducer specifications, a similar uncertainty will be associated with the result.

Figure 5 is particularly damning for non-vented transducers and we think that it needs the raw data and transducer specifications to be provided as well, or the data should not be included at all. This figure runs counter to our experience with hundreds of transducers, both vented and non-vented. It may be that you are comparing differences in full-scale, transducer type, transducer location, or perhaps the barometric compensation procedure could be improved to account for temporal offsets. Likely it is a full-scale

or transducer type issue given the smoothness of the barometric pressure data. Care needs to be taken with a figure like this to be as transparent as possible in what is being compared to not overemphasize a preference or mis-interpretation of the cause/effect.

Both vented and non-vented transducers exposed to large temperature changes will have increased uncertainty about their measurements; non-vented should not be singled out in section 6.4.1 line 21. This is more a question of deployment location (protected vs. unprotected) and less of a vented, non-vented issue. If possible, both transducer types should be deployed in protected environments that minimize the effect of the external environment while still capturing the measurements of interest (i.e., adherence to data quality objectives).

While we agree that the smaller full scale (and thus typically better accuracy and resolution) of vented transducers is a key preference and leads to some simplified calculations, the vent tube and increased cabling, particularly for even moderately deep applications, is a major downside that should be considered when selecting the optimum transducer type. We would suggest not having such a conclusive recommendation of one transducer type over the other.

**4   Other comments and specific notes**

1) We feel that there is little reason to record with such an infrequent or low monitoring frequency as 1 hour given current technology. The optimal monitoring frequency is dependent on the device hardware and the tools available for the analysis of the data. With improved tools, the hope is that monitoring at higher frequency becomes more common so that more complete water level histories are obtained. While you say that it is the maximum monitoring frequency, should this paper suggest a higher frequency of monitoring to push the profession forward?

2) The HEADCO manual by Spane 1985 also provides a thorough review of many of

the common issues related to hydraulic head measurements and should be cited as an excellent reference for the readers of this manuscript.

3) For the grouted-in application, would you still recommend vented transducers?

4) Page 2: "This is by no means trivial, and certainly much more complex than collecting manual measurements". Understanding what a manual measurement represents is also quite complex, in part because we are often missing the appropriate additional information necessary for their interpretation. Manual measurements, tend to have increased temporal uncertainty and also lack an associated barometric pressure value taken at the same time.

5) Page 4: "The hydraulic head is defined as (e.g., Freeze and Cherry, 1979)". Consider citing the original work here (Hubbert, 1940) rather than an introductory level textbook.

6) Page 6-7: "Air pressure changes act differently on the water column in open GMI than on the groundwater, because in the open GMI the air pressure change is transmitted instantaneously to the water, whereas the groundwater pressure response is more complex and can be delayed." We disagree with this statement. The air pressure changes result in formation pressure changes that are reflected in the open hole water level measurement, just perhaps at a later time period. We would argue that the water level response in an open hole would be more complex than the actual formation head. This is because open hole response contains both the formation response in addition to the responses resulting from the direct atmospheric connection and well-bore storage.

7) Page 12: "Vertical head gradients in an aquifer tend to be small under natural (i.e., not pumped) conditions, often less than $10^{-3}$". Followed by "this can be taken as an indication of the maximum head error for a typical piezometer caused by uncertainty about the elevation of the point of measurement". Given that this reasoning is used to quantify one of the forms of uncertainty based on standard practice, some basis for the gradient of $10^{-3}$ should be provided. Also, it seems very limiting to constrain

this discussion to 'aquifer' units in non-pumped systems. Larger vertical gradients should be expected across units with lower bulk vertical hydraulic conductivity and in recharge areas of a flow system or where units are being pumped (which is often the case). There are examples in the literature that show vertical gradients larger than $10^{-3}$ (see references provided in Meyer et al. 2014). Also, blending of distinct hydrostratigraphic/hydrogeologic units in a single well screen seems like an important but neglected aspect of this discussion.

8) Page 14: "In layered aquifer systems, the water level in wells with long screens was found to depend on the transmissivities of the layers intersected by the wells (Sokol, 1963)" This is the key point! This can have a dramatic influence on the head recorded even for short screens if cross-connecting the system. The measured head value becomes biased toward the highest transmissivity intersected and much of the earlier text on the monitoring point tends to confuse this issue. For a monitoring well the head is representative of the open interval and assigning the location to a point is inappropriate.

9) Page 21 Figure 6 caption: "Note that the manual dips confirm that there was no diurnal variability in the water levels (blue dots)." With the sparsity of measurements we don't think this statement is justified.

10) Page 26 Line 3: "Clock stability is an important consideration when using multiple instruments. Examples include the barometric correction of absolute pressure measurements from a non-vented transducer, or the calculation of hydraulic gradients using two different time series." Barometric correction requires the calculation of the barometric efficiency. You need barometric pressure measurements to do this calculation and therefore both non-vented and vented transducers will be affected.

11) Page 33 Line 7: Probably should site some earlier works here - for example Jacob 1940, Rojstaczer et al. 1988.

12) Page 35: "Because vented PTs measure a relative pressure instead of an absolute pressure, they have a smaller range and do not require a separate instrument to simultaneously record the atmospheric pressure." We should encourage barometric pressure to be monitored at every site. In addition, recording barometric pressure at a higher frequency can provide certain advantages related to barometric response function calculation as a more complete barometric history is obtained.

13) Page 35: "For reliably resolving head gradients and flow direction at small vertical distances, for example when assessing surface water-groundwater interactions, we recommend the use of wet/wet differential pressure sensors (e.g., Cuthbert et al., 2011)." We don't think "wet/wet differential pressure sensors" were discussed in the body of the manuscript. If not, this comment is out of place in the conclusions section. Consider adding a brief discussion to the main body of the paper or removing from the conclusions.

Thank you for this important paper,

Regards, Jonathan Kennel, Jessica Meyer, Christopher Neville, Beth Parker

Affiliations: University of Guelph, University of Iowa, S.S. Papadopulos and Associates, University of Guelph

References

Hvorslev, M.J., 1951. Time lag and soil permeability in ground-water observations.

Jacob, C.E., 1940. On the flow of water in an elastic artesian aquifer. Eos, Transactions American Geophysical Union, 21(2), pp.574-586.

Meyer, J.R., Parker, B.L., and Cherry, J.A. 2014. Characteristics of high resolution hydraulic head profiles and vertical gradients in fractured sedimentary rocks. Journal of Hydrology, 517, 493-507.

Rojstaczer, S., 1988. Determination of fluid flow properties from the response of water levels in wells to atmospheric loading. Water Resources Research, 24(11), pp.1927-

1938.

Spane Jr, F.A. and Mercer, R.B., 1985. HEADCO: A program for converting observed water levels and pressure measurements to formation pressure and standard hydraulic head (No. RHO-BW-ST–71-P). Rockwell International Corp.

---

## Referee Comment (RC1) · Donald Rosenberry (Referee) · 28 May 2019

This manuscript presents a wake-up call to the hydrogeology community that is both disturbing in its findings and exceptionally thorough and helpful. Although many of the points made throughout the manuscript are also presented in hydrogeology textbooks, they have rarely been combined in one place nor have they been researched and updated so thoroughly. Many field practitioners have been lulled into complacency by the stated accuracies of sensors capable of measuring and recording water-level data at whatever time increment is desired, forgetting that errors in manual measurements, if they are still made at all, are in addition to automated sensor errors, some of which are rarely stated or considered. The thorough listing of sources of error and their potential relative magnitudes, particularly with regard to interpretations of horizontal and vertical

hydraulic gradients, will be a useful resource.

At first, I wondered how much of an improvement this manuscript would be compared to the excellent review of this topic by Post and von Asmuth (2013). As I read further, I was impressed by the thorough coverage of sources of error, including a wide range of errors that most readers likely have not previously considered. The manuscript represents a substantial step forward in the somewhat mundane and yet very important process of collecting accurate water-level data from monitoring wells and piezometers.

The manuscript is also very well written and was a pleasure to read. Figures are clear and convey important points very convincingly. Citations to the literature are appropriate and the authors do a good job of presenting newer capabilities relative to those from decades ago.

Specific comments

Page 3, L30: Errors associated with non-vertical boreholes seem to have largely been forgotten and I was glad to see mention of this problem here. Most of the time, this error is insignificantly small. However, this error is common and, if unknown, can lead to substantial misinterpretation as the authors point out, particularly for deeper boreholes.

Page 7 L5: In addition to a lag related to barometric efficiency, many wells also suffer from a lag in the water-level response to changes in formation pressure, either because the well screen is partially clogged or improperly sized, or because the well diameter is so large that water cannot flow fast enough through the surrounding porous media and well screen to allow rapid equilibration of the water level inside the well to surrounding pressure changes (e.g.. Hvorslev, 1951). This point is not directly applicable to water-level measurement accuracy, but it could confound interpretation of barometric efficiency and could be added to this section for completeness.

Page 9: This is a very nice description of the various "accuracy" indicators and how they differ. I particularly valued the ADC component and how resolution is dependent

on sensor range. I had not seen that before.

Page 10 L24: The authors may want to mention the commonly used surveying technique of closure, or leapfrogging from a known point to unknown points and then back to the known point. Ideally, the beginning and ending locations (or elevations if surveying on the vertical axis only) will be the same, and the difference will give the user a good indication of the total-survey accuracy.

Page 12 L31-33: What saves us with regard to measuring vertical gradients is that vertical gradients tend to be much larger than horizontal gradients (both because of anisotropy and because of greater formation heterogeneity in the vertical axis, and also because piezometers designed to measure vertical gradients commonly have very short screened intervals. In the case of grouted-in piezometers, many piezometers are open only on the bottom making the screened interval essentially zero. Although the authors' points are valid, they might want to add that measurement errors can be minimized when determining vertical gradients with appropriate piezometer design and method of installation.

Page 16, L19: Here, the authors first mention use of a "dip meter" to make manual water-level measurements. One common problem with these devices is that many designs require displacement of some water before the upper sensor makes contact with the water in the well, creating an audible beep. This displacement causes minimal error for larger-diameter wells, but when the well diameter is not much larger than the diameter of the sensing device, this can cause a substantial artificial rise in the water level in the well due solely to the dip-meter measurement. Authors might want to indicate this source of error here or perhaps later in the manuscript where they talk about dip meters not having improved over the years.

Page 17 L1: The authors mention the need to re-survey well elevations in areas of unstable land surface. One source of instability that was not mentioned, but that should be included, is soil frost. Soil frost can result in vertical movement of well casing on the

order of tens of cm per year. For monitoring wells installed where the depth to the water table is small, a common occurrence near lakes or streams or wetlands, well movement due to soil frost can be a large problem that requires annual re-surveying of the altitude of the top of well casing. I have seen monitoring wells jacked completely out of the ground after several seasons. I was looking for a citation to a paper that discusses this substantial problem but the only place I found mention of this problem is in Rosenberry et al. (2008) where they write: "Shallow well casings can move vertically in response to pumping for water-sample collection, frost, and settling of well cuttings placed in the annular space between the well casing and undisturbed sediments. This is particularly common for wells installed in wetland sediments. Shallow wells constructed with plastic casing can break from ice expansion during subfreezing temperatures. Wells and surface-water staff gages located near a downwind shoreline also can be tilted, moved horizontally, or broken if surface ice is pushed onto the shoreline during fall freeze or spring thaw. Annual leveling surveys are necessary for surface-water staff gages, as well as many near-shore wells, in order to document changes in the elevation of the staff gage or the top of the well casing."

Page 20 L15: Transducer and particularly barometer error due to exposure to large temperature variation is a problem that few practitioners are aware of. An easy solution is to hang the barometer inside a well casing below ground surface but above the highest expected water level. Figure 6 provides an excellent example of the effect of allowing the barometer to be exposed to large temperature variability. However, one sentence in the figure caption is not supported by the data. The writers state in the caption, "Note that the manual dips confirm that there was no diurnal variability in the water levels (blue dots)." The manual data do not show this, nor can they. Numerous manual measurements made on the same day would be required to show diurnal response to temperature or the lack thereof. I suggest this sentence be removed from the figure caption, or perhaps be altered to indicate that the manual measurements indicated that the corrected water-level data adequately reflected the changing water level in the well.

Page 26 L16: These data regarding sensor clock drift are very disturbing. I also notice clock drift and correct for it each time I download sensor data, but I've never encountered drift this bad. I hope this is atypically bad compared to other sensors. If so, you might want to state that this table represents a perhaps extreme example.

Page 27 L9: Another source of error that has not thus far been mentioned is the offset created by displacement when a transducer is lowered into a small-diameter well that is slow to respond. If a manual measurement of depth to water is made prior to installation of the pressure transducer, output from the transducer will be related to that depth-to-water value. However, if the water level in the well rises due to displacement of water when the transducer is lowered into the well, there will be an artificial offset in the relation between transducer data and the manual water-level measurement. It is better to lower the transducer first, and then relate transducer output to a manual water-level measurement that is made at or close to the same time as a programmed sensor scan.

Page 30 Fig. 12: It took me a while to figure out what is conveyed in this very useful figure. I think a slight change to the figure caption would help lead the reader to a more efficient understanding. I suggest you revise to write "Visual comparison of horizontal and vertical random errors based on precision values (from Table 1) (note that some errors are distance dependent) for the different steps (Figure 1) and method options (Table 1)." Also, a "for example" sentence might help, either in addition to or instead of the black boxes indicated in the panels. I found it somewhat surprising that your calculated errors of about 15 percent were the same for your examples for both horizontal and vertical gradients. If you could give an example of a measured delta h, a calculated percent error based on assumed conditions and values from Table 1, and then the resulting minimum HHG or VHG, that might more clearly convey the usage of the figure.

Page 33 L2-3: The authors state that "improved standards for water level measurement would be an important step towards better hydro(geo)logical data quality and consistency." Standards have existed for many years that remain robust and are still

appropriate for modern use. One good example is from USGS (Freeman et al., 2004, p. 16). You might consider mentioning those standards as a goal that could extend more broadly throughout the hydrogeological community: "A water-level sensing and recording system should be capable of performing within a measurement error of + or – 0.01 ft. for most water-level measurement applications. For the case of large changes in water level (for example, during aquifer tests), this measurement error may not be achievable, and an accuracy of 0.1 percent of the expected range in water-level fluctuation is acceptable. Where the depth to water is greater than 100 ft, an accuracy of 0.01 percent of the estimated depth to water is generally acceptable."

Page 34 L32: Just as you stated regarding use of "dippers," it is somewhat surprising that we still are using primarily silicon strain-gage pressure transducers. Quartz-oscillator pressure transducers have been available for many years but remain little used by hydrogeologists, largely because of cost. A mention here of the exceptional accuracy of these devices might generate increased interest and demand from hydrogeologists, which may lead to larger sales and eventual reduction in unit costs.

Page 35 L1-5: I agree that vented transducers are better where their use is appropriate, but mention of the concern over keeping the vent tube unclogged and the desiccant materials fresh should also be included here. Errors resulting from improper maintenance of vented transducers can be as large as errors associated with use of a non-vented transducer and associated barometer.

Page 35 L11: Use of a transducer with a smaller pressure range to improve accuracy is another important point that often is lost. Many studies make use of transducers that have a large operating range and that are installed near the bottom of a monitoring well, when a 34 kPa transducer could be deployed at a much shallower depth with substantially greater accuracy, and for no additional cost.

Technical corrections

Page 1 L7: Change measurements to measurement.

Page 2, L 8, 22: Why do you write hydro(geo)logical with parentheses around geo? Hydrogeology is a commonly used word that is in virtually every dictionary. There is no reason for the parentheses when hydrogeology is used as an adjective.

Page 10 L12: You write, "reflecting of a target". This should be changed to "reflecting off of a target" or perhaps "reflecting from a target."

Page 15 L5: Change an to a to write "a gyroscope."

Page 16 Fig. 4: You should add titles for the x and y axes to indicate the units used. I assume they are m and mm, but you should state that for clarity. Also, I do not understand what you are conveying with the second y axis on the right side of the chart where values are listed in the order 0, 25, 50, and 5.

Page 17 L10: Consider changing unimpeded to unattended.

Page 23 Fig. 8: The y axis in panel a of Figure 8 appears to be labeled incorrectly. The axis is titled "Depth to water" but that implies that the water level inside the well (the depth to water) changed on the order of 60 to 70 m with temperature. That clearly cannot be the case. I suspect this actually is the water temperature at various depths within the standing water column inside the well. Therefore, I suggest the axis title be changed to something like "Depth below ground surface" or "Depth below water surface in well".

Page 26 Fig. 9: This is another excellent example of a common problem that all too commonly is ignored or unknown.

Page 26 L2: You might want to mention that a field laptop used for this purpose should be set to not automatically update to societally driven artificial changes in the clock, such as daylight savings time.

Page 28 Fig. 10: I assume the units on the y axis are meters because those are the units for your previous figures. However, for clarity and consistency, this should also be indicated in this figure.
Page 33 L21-22: You might add that the concern about non-vertical boreholes is a minor concern for wells that are relatively shallow. You might even include a threshold depth to water of xx m, below which most situations would result in errors that are inconsequentially small.

Page 34 L3-6: This reminder that manual measurements are still required is a very important message to convey and I was happy to see it included and emphasized in the conclusions.

References cited:

Freeman, L.A., Carpenter, M.C., Rosenberry, D.O., Rousseau, J.P., Unger, R., and McLean, J.S., 2004, Use of submersible pressure transducers in water-resources investigations: U.S. Geological Survey Techniques of Water-Resources Investigations 8-A3, 50 p. Hvorslev, M.J., 1951, Time lag and soil permeability in ground water observations: U.S. Army Corps of Engineers Waterways Experimental Station Bulletin No. 36, 50 p. Post, V.E.A., and von Asmuth, J.R., 2013, Review: Hydraulic head measurements—new technologies, classic pitfalls: Hydrogeology Journal, v. 21, no. 4, p. 737-750. Rosenberry, D.O., LaBaugh, J.W., and Hunt, R.J., 2008, Use of monitoring wells, portable piezometers, and seepage meters to quantify flow between surface water and ground water, in Rosenberry, D.O., and LaBaugh, J.W., eds., Field techniques for estimating water fluxes between surface water and ground water Denver, U.S. Geological Survey Techniques and Methods 4-D2, p. 39-70.

Referee replies to Discussion contributions

Additional contributions from Bian and Kennel et al. provide several helpful and insightful thoughts for the authors to consider. Regarding Kennel et al.'s comment no. 3 about vented versus non-vented transducers, they make a good point about not needing barometric corrections when determining horizontal gradients using multiple non-vented transducers. I also do not bother with barometric corrections when I am using two transducers to provide data related to determination of vertical gradients.

Their comment about substantial noise from the non-vented transducer in Figure 5 also raises an important point that I had missed in my review. My experiences have been similar to theirs; unless I am using a rather poor-quality transducer, I get much better response (smaller noise in the data) for barometrically corrected non-vented transducers than what is shown in Figure 5. Authors may want to provide specifications for the non-vented transducers that provided these data.

In response to Kennel et al.'s Other comments and specific notes, their question no. 3 about grouted-in applications is also a concern of mine. That situation makes me very nervous. In such an installation, we have no chance to make manual measurements once the transducer is installed. We must simply trust that the transducer is operating according to specifications. One solution is to install grouted boreholes with transducers in triplicate for each measurement installation. The authors may want to raise this consideration in their concluding remarks.

Kennel et al.'s comment/question no. 13 is also one that I had missed. Use of a wet-wet transducer is appropriate for many groundwater-surface-water installations where the need to measure vertical gradients exists and yet I see little evidence of their use in the literature. This comment is somewhat buried at the end of the concluding remarks. As Kennel et al. point out, it would be a good idea to mention the existence and special features of these transducers earlier in the manuscript.

---

## Referee Comment (RC2) · Anonymous Referee #2 · 2 Jun 2019

The authors of "Error in hydraulic head and gradient time-series measurements: a quantitative appraisal" provide interesting discussion of a fundamental concern in evaluation of field hydrogeologic data. As such, the paper has potential to make a significant contribution to the hydrologic literature. In presenting the following comments, I should note that I am principally an academic with substantial field experience. I believe that I may have approached review of this manuscript from a different viewpoint than did the other referee and the other reviews already received. I hope that this difference in viewpoint is useful to the authors.

In this vein, one overall comment that I would introduce beyond those comments already provided by Kennel and the other reviewers is that it is somewhat unclear whether this paper is intended to be a basic discussion for those working in the field

(in which case some of the additional sources of error suggested by the other reviewers might be considered) or a more theoretical analysis to help inform further study for improving estimation of hydraulic gradients and fluid fluxes in complex groundwater systems (for example, use the sources of error identified by these authors, but place them into a random numerical analysis in an effort to provide more insight into the most important errors within multiple field scenarios such as local, three-dimensional flow versus regional flow). Specifically, in reviewing this manuscript, I found that the discussion of the details of the field technologies (tapes, transducers, dip instruments) was quite fundamental (e.g., discussing the increment of measurement on a depth-to-water tape) without discussion of possible improvement, while the discussion of the magnitude of errors (and lack of discussion of interaction among errors) involved a number of assumptions. The paper has potential to be a valuable contribution, but I believe that it would benefit substantially through a bit more clarity on the intended audience (e..g, field technician versus more theory-based hydrologist) and a bit greater effort to more thoroughly understand the interplay in the identified errors. I also believe that this suggestion is reflected in some of the comments of the other reviewers (e.g., interplay of errors as suggested by Fang, the suggestions for additional types of errors by Rosenberry, the comments by Kennel et al that the example magnitude of errors should be based on a broader range of field experience and placed within context of reasonable error expectations).

More specific comments:

For many of the conclusions put forward by the authors, it might be beneficial to both suggest the implications for measurement precision in the field and avoid comparisons / generalizations that cover only a partial range of field experiences. For example, the abstract suggests that uncertainty in the hydraulic gradient "magnitude can have as great an effect on the uncertainty of flow rates as the hydraulic conductivity". I would note that these two aspects of groundwater flow analysis are fundamentally different in terms of impact on flow rates, flow direction, and response to hydraulic stimuli. I

might avoid this simplification of error comparison as it will require substantially great discussion in terms of impact on the type of final analysis desired. Further, the authors suggest that 170 meters measurement point separation is required to achieve an estimate of 10ˆ-4 in the field. Note that this implies (in perfectly one-dimensional flow) an error in differential head measurements of approximately 1.7 cm. The authors might be direct about this allowable error and briefly discuss whether this is a reasonable field result.

As noted by the other reviewers, situations in which gradients in vertical flow are of interest will often involve impact of geologic heterogeneity, natural transients (e.g., due to precipitation), and anthropogenic impacts (e.g, pumping from wells or differential densities near contamination sources). Clarification of concerns, and where those concerns are important, need to be clarified in terms of vertical gradients / vertical flow. Once again, the authors are encouraged to avoid making generalizations. This is particularly of concern in that suggestions from this manuscript might be adopted by field technical staff without careful review of the field conditions assessed by the authors and applied under conditions that are not appropriate.

I would agree with comments in one of the other reviews regarding the authors' suggestion that complexity in automated water-level measurement is significantly more complex than manual measurements. Specifically, this is perhaps inappropriate and likely ignores the complexities involved in making repeated manual measurements.

After equation 6, the authors state that del(H) is continuous. In the presence of any type of heterogeneity, this is not necessarily the case (think for example, about the instantaneous change in del(H) in the vertical direction as we move from a high K to a low K material). This statement should be corrected. More importantly, as noted in one of the other reviews, heterogeneity makes the discussion of errors in the hydraulic gradient far more complicated than even presented in this manuscript. Well geometry, well screen length and location relative to changing hydrologic units in the subsurface, screen clogging, regional variation in pumping (other wells not part of a given study),

the distance of well separation relative to the scale of heterogeneity, are all errors that make this analysis far more complex than presented here.

There is substantial concern that the authors have artificially separated "horizontal" from "vertical" gradients (e.g., equations 6-9). Certainly at the lengths scales for which errors in water levels have substantial, negative impact on our field analysis, there is no reason to make an assumption in advance that the flow field can be separated into horizontal and vertical flow and that such analysis does not vary rapidly in space. Why not base the discussion in the paper on analysis of the error in the direction and magnitude of the three-dimensional hydraulic gradient?

On page 15, the authors make some strong, sweeping conclusions about non-verticality of wells. I agree with one of the other reviewers that the authors could assist the reader by providing a bit more insight here. For example, for what minimum depth of well and in what geologic conditions will this error be most likely to impact field analysis? Further, the suggestion to use geophysical measurements to measure vertical deviation in all wells in all projects is likely beyond the financial capacity of many field efforts.

Starting on page 17, the authors make several assumptions regarding the error in the measurements of pressure transducers. As noted in at least one of the other reviews, the precision and time drift in a pressure transducer is strongly dependent on a number of factors including the type of transducer, the maximum range, and quality of construction. A bit more discussion of the range of likely precisions to be observed in the field and, as suggested by Rosenberry, careful field design can provide an opportunity to optimize field instrument design to minimize instrument errors.

I would prefer if figures 10a and 10b were presented on the same vertical scale (with some data in figure 10b shown as off range on the graph) so that the reader can actually compare the majority of the data presented.

Page 28 - I agree with the other review comment that the sampling interval of 1 hour

seems arbitrary and too long. Perhaps reconsider this suggestion.

Again, I believe that there is a potentially valuable paper presented here. I would, however, encourage the authors to consider the comments of the other reviewers as well as the comments presented here as an opportunity to substantially increase the applicability and value of the discussion presented.

————————————————————

---

## Author Response (AR2)

**Author Response to the public discussion of "*Error in hydraulic head and gradient time-series measurements: a quantitative appraisal*" by Gabriel C. Rau *et al.***

First off, we thank the Editor, reviewers and readers for their efforts in handling, reading, assessing, reviewing and commenting on our discussion article published in HESSD. We have received many positive and constructive comments during the public discussion and review. We now take the opportunity to reply to all these comments in more detail. We note that none of the comments have provided criticism that could alter the overall message. We hope that our replies will lead to the decision that we can revise the manuscript and, finally, to a consideration for publication in HESS. To make the assessment easier, we have colour coded our replies in the categories agreed (green), partially agreed (orange) and disagreed (red). Our explanations of changes to the manuscript are highlighted in blue colour.

Additional revisions that go beyond the reviewers' recommendations:
- We added to the acknowledgements: "*MAS was supported by funding from the Australian Research Council, grant DE150100302. We thank the editor Brian Berkowitz for handling, the reviewers Don Rosenberry and anonymous for assessing and some attentive readers for suggesting improvements for this work.*"
- We added a citation to Bredehoeft (1967) and McMillan et al. (2019) in appropriate places (page 21 lines 4, 5 and page 34 line 25)

Bredehoeft, J. D.: Response of well-aquifer systems to Earth tides, J. Geophys. Res., 72(12), 3075–3087, doi:10.1029/JZ072i012p03075, 1967.

McMillan, T. C., Rau, G. C., Timms, W. A. and Andersen, M. S.: Utilizing the Impact of Earth and Atmospheric Tides on Groundwater Systems: A Review Reveals the Future Potential, Rev. Geophys., 2018RG000630, doi:10.1029/2018RG000630, 2019.

**Referee Comments 1:**
This manuscript presents a wake-up call to the hydrogeology community that is both disturbing in its findings and exceptionally thorough and helpful. Although many of the points made throughout the manuscript are also presented in hydrogeology textbooks, they have rarely been combined in one place nor have they been researched and updated so thoroughly. Many field practitioners have been lulled into complacency by the stated accuracies of sensors capable of measuring and recording water-level data at whatever time increment is desired, forgetting that errors in manual measurements, if they are still made at all, are in addition to automated sensor errors, some of which are rarely stated or considered. The thorough listing of sources of error and their potential relative magnitudes, particularly with regard to interpretations of horizontal and vertical hydraulic gradients, will be a useful resource.

At first, I wondered how much of an improvement this manuscript would be compared to the excellent review of this topic by Post and von Asmuth (2013). As I read further, I was impressed by the thorough coverage of sources of error, including a wide range of errors that most readers likely have not previously considered. The manuscript represents a substantial step forward in the somewhat mundane and yet very important process of collecting accurate water-level data from monitoring wells and piezometers.

**Author Response to the public discussion of "*Error in hydraulic head and gradient time-series measurements: a quantitative appraisal*" by Gabriel C. Rau *et al.***
* * *
The manuscript is also very well written and was a pleasure to read. Figures are clear and convey important points very convincingly. Citations to the literature are appropriate and the authors do a good job of presenting newer capabilities relative to those from decades ago.

We thank the reviewer for the time taken to assess our manuscript and are pleased to receive such positive feedback. Our detailed answers are below.

Specific comments
Page 3, L30: Errors associated with non-vertical boreholes seem to have largely been forgotten and I was glad to see mention of this problem here. Most of the time, this error is insignificantly small. However, this error is common and, if unknown, can lead to substantial misinterpretation as the authors point out, particularly for deeper boreholes.

We do not see the need to revise our manuscript in response to this comment.

Page 7 L5: In addition to a lag related to barometric efficiency, many wells also suffer from a lag in the water-level response to changes in formation pressure, either because the well screen is partially clogged or improperly sized, or because the well diameter is so large that water cannot flow fast enough through the surrounding porous media and well screen to allow rapid equilibration of the water level inside the well to surrounding pressure changes (e.g.. Hvorslev, 1951). This point is not directly applicable to water-level measurement accuracy, but it could confound interpretation of barometric efficiency and could be added to this section for completeness.

We agree and will include the possibility of time lags between formation pressure and water level.

A new paragraph was inserted on page 30 in line 5 to describe time lag effects. Since it applies to multiple processes in addition to barometric effects, we chose to include the paragraph under section 6.6 Miscellaneous errors.

Page 9: This is a very nice description of the various "accuracy" indicators and how they differ. I particularly valued the ADC component and how resolution is dependent on sensor range. I had not seen that before.

Thank you. We do not see the need to revise our manuscript in response to this comment.

Page 10 L24: The authors may want to mention the commonly used surveying technique of closure, or leapfrogging from a known point to unknown points and then back to the known point. Ideally, the beginning and ending locations (or elevations if surveying on the vertical axis only) will be the same, and the difference will give the user a good indication of the total-survey accuracy.

We agree and will include a mention of leapfrogging into our revised manuscript.
* * *
**Author Response to the public discussion of "***Error in hydraulic head and gradient time-series measurements: a quantitative appraisal***" by Gabriel C. Rau *et al.***
* * *
The following sentence was inserted on page 10 line 15 "*An indication of the measurement error can be obtained by returning to the starting location of the survey and determine the difference between the recorded positions at the start and end.*"

Page 12 L31-33: What saves us with regard to measuring vertical gradients is that vertical gradients tend to be much larger than horizontal gradients (both because of anisotropy and because of greater formation heterogeneity in the vertical axis, and also because piezometers designed to measure vertical gradients commonly have very short screened intervals. In the case of grouted-in piezometers, many piezometers are open only on the bottom making the screened interval essentially zero. Although the authors' points are valid, they might want to add that measurement errors can be minimized when determining vertical gradients with appropriate piezometer design and method of installation.

We agree and will include the influence of piezometre design on the vertical gradient detection into our revised manuscript.

We have included the following sentence on page 14 line 15 "*When gradients are higher the error can be minimised by using as short a screen as possible, taking care that any pressure difference between the GMI and the formation is rapidly equilibrated by water movement (see also Section 6.6).*"

Page 16, L19: Here, the authors first mention use of a "dip meter" to make manual water-level measurements. One common problem with these devices is that many designs require displacement of some water before the upper sensor makes contact with the water in the well, creating an audible beep. This displacement causes minimal error for larger-diameter wells, but when the well diameter is not much larger than the diameter of the sensing device, this can cause a substantial artificial rise in the water level in the well due solely to the dip-meter measurement. Authors might want to indicate this source of error here or perhaps later in the manuscript where they talk about dip meters not having improved over the years.

We note that this issue is discussed in some detail in Post and von Asmuth (2013). We will add a note of this into our revised manuscript.

We replaced "*(with saltwater, for example)*" by a citation to Post and von Asmuth (2013) on page 17 line 15.

Page 17 L1: The authors mention the need to re-survey well elevations in areas of unstable land surface. One source of instability that was not mentioned, but that should be included, is soil frost. Soil frost can result in vertical movement of well casing on the order of tens of cm per year. For monitoring wells installed where the depth to the water table is small, a common occurrence near lakes or streams or wetlands, well movement due to soil frost can be a large problem that requires annual re-surveying of the altitude of the top of well casing. I have seen monitoring wells jacked completely out of the ground after several seasons. I was looking for a citation to a paper that
* * *
**Author Response to the public discussion of "*Error in hydraulic head and gradient time-series measurements: a quantitative appraisal*" by Gabriel C. Rau *et al.***

discusses this substantial problem but the only place I found mention of this problem is in Rosenberry et al. (2008) where they write: "Shallow well casings can move vertically in response to pumping for water-sample collection, frost, and settling of well cuttings placed in the annular space between the well casing and undisturbed sediments. This is particularly common for wells installed in wetland sediments. Shallow wells constructed with plastic casing can break from ice expansion during subfreezing temperatures. Wells and surface-water staff gages located near a downwind shoreline also can be tilted, moved horizontally, or broken if surface ice is pushed onto the shoreline during fall freeze or spring thaw. Annual leveling surveys are necessary for surface-water staff gages, as well as many near-shore wells, in order to document changes in the elevation of the staff gage or the top of the well casing."

Thank you. We have experienced another example of changing reference point on swelling clays in Australia. There, the whole monument was floating above the ground under dry conditions. We agree and will include these points into our revised manuscript.

We expanded this sentence into a new paragraph starting on page 17 line 20. This also accommodates the short comment by Fang Bian (see below).

Page 20 L15: Transducer and particularly barometer error due to exposure to large temperature variation is a problem that few practitioners are aware of. An easy solution is to hang the barometer inside a well casing below ground surface but above the highest expected water level. Figure 6 provides an excellent example of the effect of allowing the barometer to be exposed to large temperature variability. However, one sentence in the figure caption is not supported by the data. The writers state in the caption, "Note that the manual dips confirm that there was no diurnal variability in the water levels (blue dots)." The manual data do not show this, nor can they. Numerous manual measurements made on the same day would be required to show diurnal response to temperature or the lack thereof. I suggest this sentence be removed from the figure caption, or perhaps be altered to indicate that the manual measurements indicated that the corrected water-level data adequately reflected the changing water level in the well.

We agree and will remove the reference to confirmation from manual dips from our manuscript.

The sentence about manual dips was removed from page 21 line 20 and the caption of Figure 6 and replaced by "*Manual dips are indicated by blue dots*".

Page 26 L16: These data regarding sensor clock drift are very disturbing. I also notice clock drift and correct for it each time I download sensor data, but I've never encountered drift this bad. I hope this is atypically bad compared to other sensors. If so, you might want to state that this table represents a perhaps extreme example.

These data were acquired using standard industry loggers that are sold with the promise that clock accuracy is +/- 1 min/year. Because we have noted clock drift over the years of acquiring

groundwater hydraulic heads, we decided to test our standard loggers with the results presented in our manuscript. We believe that this is a realistic example.

We inserted "*Such deviations are unfortunately not unusual for commercial PTs (Post and von Asmuth, 2013).*" On page 27 line 15.

Page 27 L9: Another source of error that has not thus far been mentioned is the offset created by displacement when a transducer is lowered into a small-diameter well that is slow to respond. If a manual measurement of depth to water is made prior to installation of the pressure transducer, output from the transducer will be related to that depth-to-water value. However, if the water level in the well rises due to displacement of water when the transducer is lowered into the well, there will be an artificial offset in the relation between transducer data and the manual water-level measurement. It is better to lower the transducer first, and then relate transducer output to a manual water-level measurement that is made at or close to the same time as a programmed sensor scan.

We agree and will discuss the possibility that water displacement can change a measurement in our revised manuscript.

Inserted "*Observation wells may also take appreciable time to readjust after the water level inside was raised by inserting measurement instruments.*" On page 30 line 5.

Page 30 Fig. 12: It took me a while to figure out what is conveyed in this very useful figure. I think a slight change to the figure caption would help lead the reader to a more efficient understanding. I suggest you revise to write "Visual comparison of horizontal and vertical random errors based on precision values (from Table 1) (note that some errors are distance dependent) for the different steps (Figure 1) and method options (Table 1)." Also, a "for example" sentence might help, either in addition to or instead of the black boxes indicated in the panels. I found it somewhat surprising that your calculated errors of about 15 percent were the same for your examples for both horizontal and vertical gradients. If you could give an example of a measured delta h, a calculated percent error based on assumed conditions and values from Table 1, and then the resulting minimum HHG or VHG, that might more clearly convey the usage of the figure.

We agree and will revise the figure caption according to the suggestion.

We have revised the figure caption to "*Visual comparison of horizontal and vertical random errors based on precision values (from Table 1) (note that some errors are distance dependent) for the different steps (Figure 1) and method options (Table 1).*". We note that the requested example is already stated in the text (page 33 line 20). To clarify this, we added "*Please note the example error calculation in the main text.*" to the caption.

Page 33 L2-3: The authors state that "improved standards for water level measurement would be an important step towards better hydro(geo)logical data quality and consistency." Standards have existed for many years that remain robust and are still appropriate for modern use. One good

**Author Response to the public discussion of "*Error in hydraulic head and gradient time-series measurements: a quantitative appraisal*" by Gabriel C. Rau *et al.***

example is from USGS (Freeman et al., 2004, p. 16). You might consider mentioning those standards as a goal that could extend more broadly throughout the hydrogeological community: "A water-level sensing and recording system should be capable of performing within a measurement error of + or – 0.01 ft. for most water-level measurement applications. For the case of large changes in water level (for example, during aquifer tests), this measurement error may not be achievable, and an accuracy of 0.1 percent of the expected range in water-level fluctuation is acceptable. Where the depth to water is greater than 100 ft, an accuracy of 0.01 percent of the estimated depth to water is generally acceptable."

We agree and will add those goals to our manuscript alongside a reference to the source.

We have revised this paragraph as follows: "*However, the quantification and reporting of measurement error does not seem to be commonplace yet. Moreover, existing standards like Spane and Mercer (1985) or Freeman et al. (2004) contain useful guidelines for a maximum error as follows: (1) +/-3 mm (0.01 ft) for general applications, (2) 0.1 % of expected water level changes, (3) 0.01 % for cases where the depth to water exceeds ~30 m (100 ft). Such standards must see wider uptake, and the development of more sophisticated or site-specific standards, suited for a particular study area or research objective, would be important steps towards better hydrogeological data quality and consistency.*" On page 34 line 20.

Page 34 L32: Just as you stated regarding use of "dippers," it is somewhat surprising that we still are using primarily silicon strain-gage pressure transducers. Quartz oscillator pressure transducers have been available for many years but remain little used by hydrogeologists, largely because of cost. A mention here of the exceptional accuracy of these devices might generate increased interest and demand from hydrogeologists, which may lead to larger sales and eventual reduction in unit costs.

We agree and will add a mention of Quartz oscillator pressure transducers into our revised manuscript.

We inserted "*Quartz oscillator PTs are much more accurate than the commonly-used strain gauge type PT. However, they have hardly been used in groundwater studies to date, probably because of their higher cost.*" On page 36 line 25.

Page 35 L1-5: I agree that vented transducers are better where their use is appropriate, but mention of the concern over keeping the vent tube unclogged and the desiccant materials fresh should also be included here. Errors resulting from improper maintenance of vented transducers can be as large as errors associated with the use of a non-vented transducer and associated barometer.

We agree and will include a short discussion about the maintenance of vented transducers into our revised manuscript.

We inserted "*(...) , as long as there is no problem with keeping the venting tube dry.*" On page 36 line 20.

**Author Response to the public discussion of "*Error in hydraulic head and gradient time-series measurements: a quantitative appraisal*" by Gabriel C. Rau *et al.***
* * *
Page 35 L11: Use of a transducer with a smaller pressure range to improve accuracy is another important point that often is lost. Many studies make use of transducers that have a large operating range and that are installed near the bottom of a monitoring well, when a 34 kPa transducer could be deployed at a much shallower depth with substantially greater accuracy, and for no additional cost.

While we agree we caution in the case of variable density fluid inside boreholes (refer to Figure 7). Thanks for confirming our observations. We do not see a need to revise our manuscript in response to this comment.

Technical corrections
Page 1 L7: Change measurements to measurement.

We deleted the extra 's'.

Page 2, L 8, 22: Why do you write hydro(geo)logical with parentheses around geo? Hydrogeology is a commonly used word that is in virtually every dictionary. There is no reason for the parentheses when hydrogeology is used as an adjective.

We removed parentheses throughout the manuscript.

Page 10 L12: You write, "reflecting of a target". This should be changed to "reflecting off of a target" or perhaps "reflecting from a target."

We changed "of" to "off":

Page 15 L5: Change an to a to write "a gyroscope."

Changed "an" to "a".

Page 16 Fig. 4: You should add titles for the x and y axes to indicate the units used. I assume they are m and mm, but you should state that for clarity. Also, I do not understand what you are conveying with the second y axis on the right side of the chart where values are listed in the order 0, 25, 50, and 5.

Figure 4 already has titles including units. We do not understand why the reviewer did not see this. Further, the second y axis is already labelled with "Precision" and it conveys the standard deviation of the statistics shown in the first y axis. We did not see a need to add more information to this figure.

Page 17 L10: Consider changing unimpeded to unattended.

We changed „unimpeded" to „unattended".
* * *
**Author Response to the public discussion of "*Error in hydraulic head and gradient time-series measurements: a quantitative appraisal*" by Gabriel C. Rau *et al.***

Page 23 Fig. 8: The y axis in panel a of Figure 8 appears to be labeled incorrectly. The axis is titled "Depth to water" but that implies that the water level inside the well (the depth to water) changed on the order of 60 to 70 m with temperature. That clearly cannot be the case. I suspect this actually is the water temperature at various depths within the standing water column inside the well. Therefore, I suggest the axis title be changed to something like "Depth below ground surface" or "Depth below water surface in well".

We changed the y axis label to "*Depth below water surface*" and omitted "in well" because it did not fit properly.

Page 26 Fig. 9: This is another excellent example of a common problem that all too commonly is ignored or unknown.

Thank you! We made no further changes in response to this.

Page 26 L2: You might want to mention that a field laptop used for this purpose should be set to not automatically update to societally driven artificial changes in the clock, such as daylight savings time.

We agree and inserted " *(...) that is always set to the same time zone and does not update to daylight savings time.*" Page 27 line 4.

Page 28 Fig. 10: I assume the units on the y axis are meters because those are the units for your previous figures. However, for clarity and consistency, this should also be indicated in this figure.

As indicated, the units for pressure head are metres and the head gradients are dimensionless. We assumed that stating [-] would be understandable, but have changed this to [m/m] to avoid further confusion.

Page 33 L21-22: You might add that the concern about non-vertical boreholes is a minor concern for wells that are relatively shallow. You might even include a threshold depth to water of xx m, below which most situations would result in errors that are inconsequentially small.

Note that our error comparison and propagation example in Figure 12 shows that non-verticality is potentially the largest error even for a borehole that is only 10 m deep and has been logged appropriately (Figure 12a see "Point of head"). Because we consider 10 m deep wells to be classified as shallow, we wish to refrain from adding a threshold to our manuscript.

Page 34 L3-6: This reminder that manual measurements are still required is a very important message to convey and I was happy to see it included and emphasized in the conclusions.

Thank you, we are happy you share this view, too many people put too much confidence in modern electronics. We have not made any changes in response to this comment.

**Author Response to the public discussion of "*Error in hydraulic head and gradient time-series measurements: a quantitative appraisal*" by Gabriel C. Rau *et al.***

We agree with all these technical corrections and will revise our manuscript accordingly.

References cited:
Freeman, L.A., Carpenter, M.C., Rosenberry, D.O., Rousseau, J.P., Unger, R., and McLean, J.S., 2004, Use of submersible pressure transducers in water-resources investigations: U.S. Geological Survey Techniques of Water-Resources Investigations 8-A3, 50 p.

Hvorslev, M.J., 1951, Time lag and soil permeability in ground water observations: U.S. Army Corps of Engineers Waterways Experimental Station Bulletin No. 36, 50 p.

Post, V.E.A., and von Asmuth, J.R., 2013, Review: Hydraulic head measurements - new technologies, classic pitfalls: Hydrogeology Journal, v. 21, no. 4, ˇ p. 737-750.

Rosenberry, D.O., LaBaugh, J.W., and Hunt, R.J., 2008, Use of monitoring wells, portable piezometers, and seepage meters to quantify flow between surface water and ground water, in Rosenberry, D.O., and LaBaugh, J.W., eds., Field techniques for estimating water fluxes between surface water and ground water Denver, U.S. Geological Survey Techniques and Methods 4-D2, p. 39-70.

We will add these literature suggestions as citations in appropriate places to our manuscript.

We have added these citations in accordance with our revisions made in response to the reviewers' comments.

Referee replies to Discussion contributions

Additional contributions from Bian and Kennel et al. provide several helpful and insightful thoughts for the authors to consider. Regarding Kennel et al.'s comment no. 3 about vented versus non-vented transducers, they make a good point about not needing barometric corrections when determining horizontal gradients using multiple non-vented transducers. I also do not bother with barometric corrections when I am using two transducers to provide data related to determination of vertical gradients. Their comment about substantial noise from the non-vented transducer in Figure 5 also raises an important point that I had missed in my review. My experiences have been similar to theirs; unless I am using a rather poor-quality transducer, I get much better response (smaller noise in the data) for barometrically corrected non-vented transducers than what is shown in Figure 5. Authors may want to provide specifications for the non-vented transducers that provided these data.

Our data is based on a cheap pressure transducer which shows more noise than the others. We will make sure to mention this so that the reader does not walk away with the impression that non-vented transducers are inferior.

We inserted "*The low precision achieved by the non-vented PT is specific for the particular instrument used and not representative for all PTs of this type.*" in the caption of Figure 5.

**Author Response to the public discussion of "*Error in hydraulic head and gradient time-series measurements: a quantitative appraisal*" by Gabriel C. Rau *et al.***

In response to Kennel et al.'s Other comments and specific notes, their question no. 3 about grouted-in applications is also a concern of mine. That situation makes me very nervous. In such an installation, we have no chance to make manual measurements once the transducer is installed. We must simply trust that the transducer is operating according to specifications. One solution is to install grouted boreholes with transducers in triplicate for each measurement installation. The authors may want to raise this consideration in their concluding remarks.

We agree and will include a short discussion about the risks of grouted in piezos as well as the need for redundancy with regards to measurement instruments.

Inserted "*One strategy to verify PT performance in that case is to install three instruments at the same depth.*" On page 18 line 34.

Kennel et al.'s comment/question no. 13 is also one that I had missed. Use of a wet-wet transducer is appropriate for many groundwater-surface-water installations where the need to measure vertical gradients exists and yet I see little evidence of their use in the literature. This comment is somewhat buried at the end of the concluding remarks. As Kennel et al. point out, it would be a good idea to mention the existence and special features of these transducers earlier in the manuscript.

We agree and will discuss the advantages of wet/wet pressure transducers in our revised manuscript

We have added a short explanation of the usefulness of wet/wet pressure transducers starting on page 19 line 4: "*Wet/wet pressure transducers measure the pressure difference between two points that are both exposed to water (Cuthbert et al., 2011). Such devices are ideal for obtaining small head gradients, such as is required for measuring groundwater-surface water interactions, because they eliminate the uncertainties arising from barometric correction or the spatial positioning of two individual measurement points.*"

**Referee Comments 2:**
The authors of "Error in hydraulic head and gradient time-series measurements: a quantitative appraisal" provide interesting discussion of a fundamental concern in evaluation of field hydrogeologic data. As such, the paper has potential to make a significant contribution to the hydrologic literature. In presenting the following comments, I should note that I am principally an academic with substantial field experience. I believe that I may have approached review of this manuscript from a different viewpoint than did the other referee and the other reviews already received. I hope that this difference in viewpoint is useful to the authors.

We thank the reviewer for her/his valuable assessment.

In this vein, one overall comment that I would introduce beyond those comments already provided by Kennel and the other reviewers is that it is somewhat unclear whether this paper is intended to be a basic discussion for those working in the field (in which case some of the additional sources of error suggested by the other reviewers might be considered) or a more theoretical analysis to help inform

**Author Response to the public discussion of "*Error in hydraulic head and gradient time-series measurements: a quantitative appraisal*" by Gabriel C. Rau *et al.***

further study for improving estimation of hydraulic gradients and fluid fluxes in complex groundwater systems (for example, use the sources of error identified by these authors, but place them into a random numerical analysis in an effort to provide more insight into the most important errors within multiple field scenarios such as local, three-dimensional flow versus regional flow). Specifically, in reviewing this manuscript, I found that the discussion of the details of the field technologies (tapes, transducers, dip instruments) was quite fundamental (e.g., discussing the increment of measurement on a depth-to-water tape) without discussion of possible improvement, while the discussion of the magnitude of errors (and lack of discussion of interaction among errors) involved a number of assumptions. The paper has potential to be a valuable contribution, but I believe that it would benefit substantially through a bit more clarity on the intended audience (e..g, field technician versus more theory-based hydrologist) and a bit greater effort to more thoroughly understand the interplay in the identified errors. I also believe that this suggestion is reflected in some of the comments of the other reviewers (e.g., interplay of errors as suggested by Fang, the suggestions for additional types of errors by Rosenberry, the comments by Kennel et al that the example magnitude of errors should be based on a broader range of field experience and placed within context of reasonable error expectations).

Our target audience is the groundwater community at large. We believe that our review will benefit those that collect and use field data as it raises awareness about potential sources of errors, some of which are often overlooked, as noted by the first reviewer. But at the same time we think that this paper is equally relevant for academics and theoretical groundwater modellers, even if they never collect any field data themselves. Hydraulic head data is frequently taken from public databases or other third-party sources to be used in model calibration and without in-depth understanding of the causes and magnitudes of measurement errors, the limitations posed by the data accuracy may not be fully appreciated by the user(s).

We note that the scientific objectives determine the required accuracy. Hence, scientists must be as aware of the operating procedures as the field practitioners who make the measurements. Otherwise, they cannot design their research. Essentially, the paper is written for both cases but mainly for academics. The aspects that we highlight are often unknown or ignored which leads to bad data (and thus conclusions). We hope that this aim fulfils the reviewer's expectations.

In response to the point about the interplay between systematic errors, we have rephrase the sentence on page 3 line 7 to "*Unrecognised and unaccounted for systematic errors can accumulate or cancel, leading to unquantifiable inaccuracies.*"

More specific comments:
For many of the conclusions put forward by the authors, it might be beneficial to both suggest the implications for measurement precision in the field and avoid comparisons / generalizations that cover only a partial range of field experiences. For example, the abstract suggests that uncertainty in the hydraulic gradient "magnitude can have as great an effect on the uncertainty of flow rates as the hydraulic conductivity". I would note that these two aspects of groundwater flow analysis are fundamentally different in terms of impact on flow rates, flow direction, and response to hydraulic

**Author Response to the public discussion of "*Error in hydraulic head and gradient time-series measurements: a quantitative appraisal*" by Gabriel C. Rau *et al.***

stimuli. I might avoid this simplification of error comparison as it will require substantially great discussion in terms of impact on the type of final analysis desired. Further, the authors suggest that 170 meters measurement point separation is required to achieve an estimate of 10ˆ-4 in the field. Note that this implies (in perfectly one-dimensional flow) an error in differential head measurements of approximately 1.7 cm. The authors might be direct about this allowable error and briefly discuss whether this is a reasonable field result.

We respectfully disagree. Our analysis clearly shows that the accuracy of the head change is dependent on the distance between the measurement points. While it is impossible to measure 1.7 cm over 10 km, it might work better over a distance of 10 m. Therefore, casting the discussion in terms of gradients is much more useful than head differences.

Note that head measurements are made and interpreted regardless of system heterogeneity, transience or anthropogenic impact. Errors in head measurements will be present and affect groundwater flow estimates under all conditions. Therefore, minimising the error by adhering to our recommendations would be of interest in all situations. In contrast, one could argue that minimising head measurement errors could enable improved analysis of heterogeneity. To further clarify the intent of our manuscript, we:
- Changed the criticised sentence in the abstract to "*There is sufficient information in the literature to suggest that head measurement errors can impede the reliable detection of flow directions and significantly increase the uncertainty of groundwater flow rate calculations.*"
- We inserted "*We acknowledge that quantifying groundwater flow requires knowledge of the distribution of hydraulic conductivity in addition to hydraulic gradients. While this can be highly heterogeneous and could further complicate investigations, we focus on minimising hydraulic head and gradient measurement errors because doing so increases the accuracy of flow estimates or hydraulic property inversions.*" Starting on page 3 line 21.
- We further inserted "*In practice, heterogeneity of the hydraulic conductivity will further add to the uncertainty of groundwater flow estimates.*" Starting on page 33 line 26.

As noted by the other reviewers, situations in which gradients in vertical flow are of interest will often involve impact of geologic heterogeneity, natural transients (e.g., due to precipitation), and anthropogenic impacts (e.g, pumping from wells or differential densities near contamination sources). Clarification of concerns, and where those concerns are important, need to be clarified in terms of vertical gradients / vertical flow. Once again, the authors are encouraged to avoid making generalizations. This is particularly of concern in that suggestions from this manuscript might be adopted by field technical staff without careful review of the field conditions assessed by the authors and applied under conditions that are not appropriate.

We agree and will include discussion of realistic gradients into our manuscript. Please note that our generalisations are required in order for the complexity of this topic to be simplified to the reader. We will aim to include appropriate caveats wherever necessary.

We note that our revisions from the previous comment also covers the current recommendation.

**Author Response to the public discussion of "*Error in hydraulic head and gradient time-series measurements: a quantitative appraisal*" by Gabriel C. Rau *et al.***

I would agree with comments in one of the other reviews regarding the authors' suggestion that complexity in automated water-level measurement is significantly more complex than manual measurements. Specifically, this is perhaps inappropriate and likely ignores the complexities involved in making repeated manual measurements.

In our manuscript we say that the technology involved in making automated measurements is more complex. This includes the data processing, for which automated measurements and QA are much more complicated compared to manual measurements. We hope that our manuscript reflects this viewpoint accurately.

We have not made any revisions in response to this comment.

After equation 6, the authors state that del(H) is continuous. In the presence of any type of heterogeneity, this is not necessarily the case (think for example, about the instantaneous change in del(H) in the vertical direction as we move from a high K to a low K material). This statement should be corrected. More importantly, as noted in one of the other reviews, heterogeneity makes the discussion of errors in the hydraulic gradient far more complicated than even presented in this manuscript. Well geometry, well screen length and location relative to changing hydrologic units in the subsurface, screen clogging, regional variation in pumping (other wells not part of a given study), the distance of well separation relative to the scale of heterogeneity, are all errors that make this analysis far more complex than presented here.

While there may be rapid head changes in space, we believe that true head discontinuities are rare (with the exception perhaps of seepage faces). We note that the focus of our manuscript is explicitly on hydraulic heads and not on geological heterogeneities. We will include this as a caveat into our revised manuscript.

We believe that our revisions to an earlier comment made by this reviewer sufficiently cover revisions required for this comment. See the new statements in the introduction (page 3 line 21) and conclusion (page 33 line 26) which further clarify this and delineate the focus of our manuscript.

There is substantial concern that the authors have artificially separated "horizontal" from "vertical" gradients (e.g., equations 6-9). Certainly at the lengths scales for which errors in water levels have substantial, negative impact on our field analysis, there is no reason to make an assumption in advance that the flow field can be separated into horizontal and vertical flow and that such analysis does not vary rapidly in space. Why not base the discussion in the paper on analysis of the error in the direction and magnitude of the three-dimensional hydraulic gradient?

We do not understand the difference between what we have done and what the reviewer requests us to do. We have merely broken down a 3D flow field into its cartesian components, horizontal and vertical. Many measurement techniques require separate horizontal and vertical errors. This is

**Author Response to the public discussion of "*Error in hydraulic head and gradient time-series measurements: a quantitative appraisal*" by Gabriel C. Rau *et al.***

standard practise in groundwater investigations and modelling. We do not see the need to revise our manuscript in response to this comment.

We note also that we specifically mention "*(..), it is rare for field studies to determine $\nabla h$ in three dimensions.*" On page 6 in line 15.

On page 15, the authors make some strong, sweeping conclusions about nonverticality of wells. I agree with one of the other reviewers that the authors could assist the reader by providing a bit more insight here. For example, for what minimum depth of well and in what geologic conditions will this error be most likely to impact field analysis? Further, the suggestion to use geophysical measurements to measure vertical deviation in all wells in all projects is likely beyond the financial capacity of many field efforts.

We were surprised to see very little literature regarding this topic. We use published statistics to show that non-verticality is a serious issue that has been neglected by the hydrogeological community. In our error example, we use a realistic deviation and a small well depth to show just how large the error from non-verticality can be. We therefore disagree that our statements are sweeping and refer to RC1 who specifically appreciates us raising this issue. We agree that there are serious financial implications, and certainly it may not always be feasible to fulfill this recommendation, but in that case the uncertainty should be acknowledged and an assessment of the potential error included.

We have not made any revisions in response to this comment.

Starting on page 17, the authors make several assumptions regarding the error in the measurements of pressure transducers. As noted in at least one of the other reviews, the precision and time drift in a pressure transducer is strongly dependent on a number of factors including the type of transducer, the maximum range, and quality of construction. A bit more discussion of the range of likely precisions to be observed in the field and, as suggested by Rosenberry, careful field design can provide an opportunity to optimize field instrument design to minimize instrument errors.

We will try to include more discussion about factors that degrade the precision of pressure transducers into our revised manuscript. We had explored the idea of providing a comprehensive list of the most common manufactures and pressure transducer types with reported specs, however, this was beyond the scope of this paper. It would however, be a valuable follow up to this study.

After careful consideration we have refrained from making substantial changes to the manuscript because adding a "range of likely precisions" as requested by the reviewer proved impossible due to the many factors that play a role. We like to point out that we already provided an estimate of the best achievable performance (deviations between sensors on the order of mm over a 15 month measurement period) in the last paragraph on page 17 of the original manuscript. In response to this comment we inserted "*(e.g., within a few centimetres)*" on page 34 line 15 to indicate the best achievable accuracy based on our own experience.

**Author Response to the public discussion of "*Error in hydraulic head and gradient time-series measurements: a quantitative appraisal*" by Gabriel C. Rau *et al.***

I would prefer if figures 10a and 10b were presented on the same vertical scale (with some data in figure 10b shown as off range on the graph) so that the reader can actually compare the majority of the data presented.

We agree and will make the scales in Figures 10a and 10b equal.

We have changed the extents of the y axes in Figure 10 to equal.

Page 28 - I agree with the other review comment that the sampling interval of 1 hour seems arbitrary and too long. Perhaps reconsider this suggestion.

An optimal sampling interval is a controversial issue. Here, a balance has to be found between generating too much data for groundwater responses that are slow and missing details for dynamic systems. We will attempt to discuss this in a bit more detail in our revised manuscript.

We rephrased the sentences starting on page 29 line 8 to "*To avoid such errors, a suitable measurement interval must be chosen upon initial logger deployment, which depends on the hydrogeological conditions at the measurement location. Only when it becomes clear that there is no temporal variability at this timescale can the sampling interval be increased to avoid unnecessary data handling and storage requirements.*"

Again, I believe that there is a potentially valuable paper presented here. I would, however, encourage the authors to consider the comments of the other reviewers as well as the comments presented here as an opportunity to substantially increase the applicability and value of the discussion presented.

We thank the reviewer and will do our best to address all the comments received during the review process.

**Short Comments 1:**
Hello: First of all, thanks for the precious recommendations for minimising the systematic errors. It's quite practical but not many people ever considered. I've three aspects to ask:

1) Change on the surface elevation: It's generally not considered for the most case, but there's a special occasion, that we recorded a surface subsidence from a 8 m GMI due to the severe drought in Thirlmere Lakes, NSW, Australia. It's been noticed because it's so visible, that without any changes on the GMI, the water level suspiciously increased for 3 days around 6 pm for 3 days. Then no more changes were monitored. Every change lasts around 1 hour with more than 100 mm increase. I think this phenomenon might associate with temperature effects or simply aquifer changes. Not sure if there's good method to mitigate?

We agree and will include the possibility of a change in reference point into our revised manuscript (together with the issue of freezing conditions raised by reviewer 1).

**Author Response to the public discussion of "*Error in hydraulic head and gradient time-series measurements: a quantitative appraisal*" by Gabriel C. Rau *et al.***

We expanded the discussion of the contributing factors to land surface movement. It starts on page 17 line 20 with "*The causes for such movements (...)*"

2) Change on the logger position from the wire connected: It's been mentioned, but not fully discussed in the main content. It mainly occurs with a vented logger and the venting cable, which has larger diameters than regular wires. Especially when the piezometer is narrow, the changes from that might be a big problem for this bore during a relatively large water recharge/pumping. That's why I personally don't recommend vented logger.

We agree and will briefly discuss the possibility that a transducer can change its vertical position in our revised manuscript.

Inserted "*The cables of vented PTs may be large relative to the well diameter, and sometimes there is little room for the desiccation unit at the top, which may mean that the logger is not always returned to exactly the same position after the GMI was accessed for maintenance or other measurements (e.g. water sampling).*" Starts on page 26 line 12.

3) As some errors might offset other errors, and we already have a brief idea for some general ranges of errors, is that possible to have an universal accepted errors for the whole records?

We agree and point out that this is an extremely complicated issue. Once multiple errors are superimposed it is likely impossible to disentangle the individual effects. Therefore, it is extremely important to understand which systematic errors can occur, so that they can be identified and eliminated as much as possible. This avoids the superposition problem, at least partially. We will mention this in our revised manuscript.

We inserted "*It is difficult to establish if systematic errors are accumulating or cancelling, and hence they must be avoided, or identified and corrected.*" On page 34 line 7.

Regards, Fang

**Short Comments 2:**
Comments by: Jonathan Kennel, Jessica Meyer, Christopher Neville, Beth Parker

This paper provides a good review of many of the factors that can influence hydraulic head measurements and it is nice to see increased focus on such a fundamental measurement. There are three areas in the paper where we feel further clarification is warranted: time-lag of the monitoring well; calculation of hydraulic gradient; and the comparison of vented and non-vented transducers. Our comments address relatively subtle points and not intended to detract from the overall emphasis of the paper.

**Author Response to the public discussion of "*Error in hydraulic head and gradient time-series measurements: a quantitative appraisal*" by Gabriel C. Rau *et al.***

Many thanks for your positive assessment, and for taking the time to provide the valuable comments below.

1 Time-lag

It should also be emphasized that while the equivalent freshwater water level (corrections for density) or equivalent piezometric level (for shut in measurements) are often used to infer flow directions and calculate hydraulic gradients, the water level elevation in an open well is not the fundamental parameter of interest. The hydraulic head in the formation is what drives flow. A key distinction is that often a monitoring well or piezometer requires groundwater flow between the formation and monitoring well to record a change in water level surface. This flow is not instantaneous and is commonly referred to as time-lag (Hvorslev, 1951). This time-lag is different than the timelag associated with the propagation of barometric pressure through the vadose zone mentioned in the current paper. The length of the time-lag is dependent on well-bore storage and formation properties; the water level measured from an open monitoring device (or even a grouted transducer to a much smaller extent) will incorporate the effects of antecedent changes in formation pressure. Monitoring wells do not all have the same time-lag associated with them, which also adds temporal uncertainty when comparing measurements. As monitoring frequency increases the uncertainty associated with variable time-lags will become more important and apparent.

We agree and will discuss the possibility of time lags between formation pressure and water levels into our revised manuscript. This point was also raised by reviewer 1.

We inserted a new paragraph to address this issue. It starts on page 30 line 1 with "*Open GMI may suffer from a lag in the water-level response (...)*"

2 Hydraulic gradients

The locations of the monitoring wells play a very important role in the calculation of the hydraulic gradient, not only the knowledge of their true position (x and y, z, and time coordinates) but how the wells are oriented in relation to each other. For example, it is worth mentioning that for calculating a planar hydraulic gradient from three wells in the same hydrogeologic unit the optimal arrangement is in the form of an equilateral triangle. As the locations deviate further from an equilateral triangle the uncertainty of water level measurements plays an increasingly important role in gradient calculations. Gradients vary in space and time, so with increasing monitoring distances the spatial confidence of the calculated gradient actually declines (i.e., uncertainty around the representativeness of the gradient). We are limited by our devices but we strive to get gradients across appropriate scales. It should be mentioned in the discussion on vertical gradients that avoiding blending of distinct hydrogeologic/hydrostratigraphic units is critical to accurately calculating meaningful vertical gradients.

We agree and will include a brief discussion of the influence of screen position of the calculation of gradients into our revised manuscript.

**Author Response to the public discussion of "*Error in hydraulic head and gradient time-series measurements: a quantitative appraisal*" by Gabriel C. Rau *et al.***

We inserted "*(...), which are best arranged in the form of an equilateral triangle, (...)*" on page 6 line 27 and "*For accurate vertical gradients it is important to use short screens that are within a single hydrogeological unit.*" On page 7 line 2.

**3 Vented and non-vented transducers**

One of the main arguments against non-vented transducers is that you need to convert these values to an open hole water level measurement. While this may be the case when you want to compare the value to a manual measurement in a conventional piezometer, in many cases this is not necessary. For example, if you are trying to calculate a horizontal gradient between three wells in the same aquifer, and all of the transducers deployed are non-vented, there is no need to first remove the barometric component from the results if the elevations of the sensor of each transducer are known. Converting to an equivalent water level may just add uncertainty that is not necessary. Another example is for calculating barometric/loading efficiency. The main issue is related to time shift between equipment, which affects both methods similarly. Given the same transducer specifications, a similar uncertainty will be associated with the result. Figure 5 is particularly damning for non-vented transducers and we think that it needs the raw data and transducer specifications to be provided as well, or the data should not be included at all. This figure runs counter to our experience with hundreds of transducers, both vented and non-vented. It may be that you are comparing differences in full-scale, transducer type, transducer location, or perhaps the barometric compensation procedure could be improved to account for temporal offsets. Likely it is a full-scale or transducer type issue given the smoothness of the barometric pressure data. Care needs to be taken with a figure like this to be as transparent as possible in what is being compared to not overemphasize a preference or mis-interpretation of the cause/effect. Both vented and non-vented transducers exposed to large temperature changes will have increased uncertainty about their measurements; non-vented should not be singled out in section 6.4.1 line 21. This is more a question of deployment location (protected vs. unprotected) and less of a vented, non-vented issue. If possible, both transducer types should be deployed in protected environments that minimize the effect of the external environment while still capturing the measurements of interest (i.e., adherence to data quality objectives). While we agree that the smaller full scale (and thus typically better accuracy and resolution) of vented transducers is a key preference and leads to some simplified calculations, the vent tube and increased cabling, particularly for even moderately deep applications, is a major downside that should be considered when selecting the optimum transducer type. We would suggest not having such a conclusive recommendation of one transducer type over the other.

We agree that we should be careful to make recommendations about transducer types, and indeed there are other factors at play that determine a transducer's performance. What we meant to illustrate in Figure 5b was that not just any logger should be used to measure water level changes. This may seem like an obvious point, but we have noted all too often that the choice for an instrument is made based on practical and logistical considerations (e.g., availability, affordability, etc.) rather than scientific objectives. We will strengthen this message, and will make sure that we do not provide a recommendation of vented over non-vented based on false comparisons.

**Author Response to the public discussion of "*Error in hydraulic head and gradient time-series measurements: a quantitative appraisal*" by Gabriel C. Rau *et al.***

In addition to the changes in response to a number of suggestions by reviewer #1 that were similar to this one, we included the following sentence on page 21 line 8 "*It should be noted that when PTs are used to calculate gradients, the readings from non-vented PTs may be used directly without compensating for atmospheric pressure changes as long as the PTs all experience the same atmospheric pressure change.*"

4 Other comments and specific notes

1) We feel that there is little reason to record with such an infrequent or low monitoring frequency as 1 hour given current technology. The optimal monitoring frequency is dependent on the device hardware and the tools available for the analysis of the data. With improved tools, the hope is that monitoring at higher frequency becomes more common so that more complete water level histories are obtained. While you say that it is the maximum monitoring frequency, should this paper suggest a higher frequency of monitoring to push the profession forward?

We believe that monitoring should start at the highest feasible frequency until the dynamics of a system are revealed. After that there is the potential to reduce sampling intervals. There is a trade-off between generating too much data in deep (static) and missing the water level history in a shallow (dynamic) groundwater system. We will clarify this in our revised manuscript.

We rephrased the sentence on page 29 line 9 to "*To avoid such errors, a suitable measurement interval must be chosen upon initial logger deployment, which depends on the hydrogeological conditions at the measurement location. Only when it becomes clear that there is no temporal variability at this timescale can the sampling interval be increased to avoid unnecessary data storage requirements.*"

2) The HEADCO manual by Spane 1985 also provides a thorough review of many of the common issues related to hydraulic head measurements and should be cited as an excellent reference for the readers of this manuscript.

We agree and will cite this reference in appropriate places in our revised manuscript.

We inserted a reference to Spane (1985) on page 34 in line 18.

3) For the grouted-in application, would you still recommend vented transducers?

That question cannot be answered simply with yes or no because it depends on the case. We do not see a need to revise our manuscript in response to this question.

4) Page 2: "This is by no means trivial, and certainly much more complex than collecting manual measurements". Understanding what a manual measurement represents is also quite complex, in part because we are often missing the appropriate additional information necessary for their interpretation. Manual measurements, tend to have increased temporal uncertainty and also lack an associated barometric pressure value taken at the same time.

**Author Response to the public discussion of "*Error in hydraulic head and gradient time-series measurements: a quantitative appraisal*" by Gabriel C. Rau *et al.***
* * *
The purpose of manual measurements is to determine the water level in the monitoring bore and to transform automated time series into pressure heads. This can subsequently be used to conduct barometric corrections using an appropriate pressure record. We believe that this is sufficiently covered in our manuscript.

5) Page 4: "The hydraulic head is defined as (e.g., Freeze and Cherry, 1979)". Consider citing the original work here (Hubbert, 1940) rather than an introductory level textbook.

We agree and will cite the provided reference in our revised manuscript. Note that we already cited this textbook for the reader's convenience (especially since it has become freely available online).

We added the citation Hubbert (1940) on page 4 line 4.

6) Page 6-7: "Air pressure changes act differently on the water column in open GMI than on the groundwater, because in the open GMI the air pressure change is transmitted instantaneously to the water, whereas the groundwater pressure response is more complex and can be delayed." We disagree with this statement. The air pressure changes result in formation pressure changes that are reflected in the open hole water level measurement, just perhaps at a later time period. We would argue that the water level response in an open hole would be more complex than the actual formation head. This is because open hole response contains both the formation response in addition to the responses resulting from the direct atmospheric connection and well-bore storage.

Our statement is in principal correct when you consider a semi-confined systems which is poroelastic. We disagree with your statement "The air pressure changes result in formation pressure changes that are reflected in the open hole water level measurement, just perhaps at a later time period". This is because the formation pressure changes will depend on the barometric efficiency. We do agree with your last statement and will clarify this in our revised manuscript.

We changed our formulation to "*Air pressure changes are transmitted instantaneously to the water column in open GMI. In contrast, the formation response is more complex because air pressure changes must propagation through the subsurface to the point of measurement which can result in a delay.*" On page 7 line 5.

7) Page 12: "Vertical head gradients in an aquifer tend to be small under natural (i.e., not pumped) conditions, often less than $10^{-3}$ ". Followed by "this can be taken as an indication of the maximum head error for a typical piezometer caused by uncertainty about the elevation of the point of measurement". Given that this reasoning is used to quantify one of the forms of uncertainty based on standard practice, some basis for the gradient of $10^{-3}$ should be provided. Also, it seems very limiting to constrain this discussion to 'aquifer' units in non-pumped systems. Larger vertical gradients should be expected across units with lower bulk vertical hydraulic conductivity and in recharge areas of a flow system or where units are being pumped (which is often the case). There are examples in the literature that show vertical gradients larger than $10^{-3}$ (see references provided in Meyer et al.

2014). Also, blending of distinct hydrostratigraphic/hydrogeologic units in a single well screen seems like an important but neglected aspect of this discussion.

The basis for this value is the upper end of the range of recharge rates (1 mm/d) for a K = 1 m/d. We will clarify this in our revised manuscript.

We inserted "*(...) (this value would be typical for an aquifer with a rainfall recharge rate of 1 mm d$^{-1}$ and a vertical hydraulic conductivity of 1 m d$^{-1}$)*" page 14 line 8.

8) Page 14: "In layered aquifer systems, the water level in wells with long screens was found to depend on the transmissivities of the layers intersected by the wells (Sokol, 1963)" This is the key point! This can have a dramatic influence on the head recorded even for short screens if cross-connecting the system. The measured head value becomes biased toward the highest transmissivity intersected and much of the earlier text on the monitoring point tends to confuse this issue. For a monitoring well the head is representative of the open interval and assigning the location to a point is inappropriate.

We agree and have tried to make this point clear. We will further clarify this in our revised manuscript.

We inserted "*For accurate vertical gradients it is important to use short screens that are within a single hydrogeological unit.*" Page 7 line 2.

9) Page 21 Figure 6 caption: "Note that the manual dips confirm that there was no diurnal variability in the water levels (blue dots)." With the sparsity of measurements we don't think this statement is justified.

We agree that this is not clear, and will remove this statement. This was also raised by reviewer 1.

The sentence about manual dips was removed from page 21 line 23 and the caption of Figure 6.

10) Page 26 Line 3: "Clock stability is an important consideration when using multiple instruments. Examples include the barometric correction of absolute pressure measurements from a non-vented transducer, or the calculation of hydraulic gradients using two different time series." Barometric correction requires the calculation of the barometric efficiency. You need barometric pressure measurements to do this calculation and therefore both non-vented and vented transducers will be affected.

We agree and will clarify this point in our revised manuscript.

...

11) Page 33 Line 7: Probably should site some earlier works here - for example Jacob 1940, Rojstaczer et al. 1988.

We agree and will cite these works in our revised manuscript.

We inserted a citation to Jacob (1940) page 7 line 28.

12) Page 35: "Because vented PTs measure a relative pressure instead of an absolute pressure, they have a smaller range and do not require a separate instrument to simultaneously record the atmospheric pressure." We should encourage barometric pressure to be monitored at every site. In addition, recording barometric pressure at a higher frequency can provide certain advantages related to barometric response function calculation as a more complete barometric history is obtained.

We agree that barometric data is very important and will clarify this statement in our revised manuscript.

We inserted "*Nevertheless, barometric pressure must still be acquired in order to perform a barometric correction.*" Page 36 line 23.

13) Page 35: "For reliably resolving head gradients and flow direction at small vertical distances, for example when assessing surface water-groundwater interactions, we recommend the use of wet/wet differential pressure sensors (e.g., Cuthbert et al., 2011)." We don't think "wet/wet differential pressure sensors" were discussed in the body of the manuscript. If not, this comment is out of place in the conclusions section. Consider adding a brief discussion to the main body of the paper or removing from the conclusions.

We agree and will include some more discussion of wet/wet pressure transducers into our revised manuscript. This was also raised by reviewer 1.

We have inserted "*Wet/wet pressure transducers measure the pressure difference between two points that are both exposed to water \citep{Cuthbert2011}. Such devices are ideal for obtaining small head gradients, such as is required for measuring groundwater-surface water interactions, because they eliminate the uncertainties arising from barometric correction or the spatial positioning of two individual measurement points.*" Page 19 line 5.

Thank you for this important paper,
Regards, Jonathan Kennel, Jessica Meyer, Christopher Neville, Beth Parker

Affiliations: University of Guelph, University of Iowa, S.S. Papadopulos and Associates, University of Guelph References

References
Hvorslev, M.J., 1951. Time lag and soil permeability in ground-water observations.

Jacob, C.E., 1940. On the flow of water in an elastic artesian aquifer. Eos, Transactions American Geophysical Union, 21(2), pp.574-586.

Meyer, J.R., Parker, B.L., and Cherry, J.A. 2014. Characteristics of high resolution hydraulic head profiles and vertical gradients in fractured sedimentary rocks. Journal of Hydrology, 517, 493-507.

Rojstaczer, S., 1988. Determination of fluid flow properties from the response of water levels in wells to atmospheric loading. Water Resources Research, 24(11), pp.1927- 1938.

Spane Jr, F.A. and Mercer, R.B., 1985. HEADCO: A program for converting observed water levels and pressure measurements to formation pressure and standard hydraulic head (No. RHO-BW-ST–71-P). Rockwell International Corp.

We will include the suggested references at appropriate places in our revised manuscript.